# MULTI-MODAL CONTROLLED COHERENT MOTION SYNTHESIS

## ABSTRACT

We walk and talk at the same time all the time. It is just natural for us. This paper tackles the challenge of replicating such natural behaviors in 3D avatar motion generation driven by concurrent multi-modal inputs, e.g., a text description "a man is walking" alongside a speech audio. Existing methods, constrained by the scarcity of aligned multi-modal data, typically combine motions from individual modalities sequentially or through weighted averaging. These strategies often result in mismatched or unrealistic movements. To overcome these limitations, we propose **MOCO**, a novel diffusion-based framework capable of processing multiple simultaneous inputs—including speech audio, text descriptions, and trajectory data—to generate coherent and lifelike motions without requiring additional datasets. Our key innovation lies in decoupling the motion generation process. During each denoising step, the diffusion model independently generates motions for each modality from the input noise and assembles the body parts according to predefined spatial rules. The resulting combined motion is then diffused and serves as the input noise for the subsequent denoising step. This iterative approach enables each modality to refine its contribution within the context of the overall motion, progressively harmonizing movements across modalities. Consequently, the generated motions become increasingly natural and fluid with each iteration, achieving coherent and synchronized behaviors. We evaluate our approach using a purpose-built multi-modal benchmark. Experimental results demonstrate that **MOCO** significantly outperforms existing baselines, advancing the field of multi-modal motion generation for 3D avatars. The code will be released.

## 1 INTRODUCTION

Imagine watching a virtual talk show where the host delivers engaging dialogue complemented by expressive gestures, natural body movements, and precise movement paths. The host walks across the stage following a scripted trajectory, uses hand gestures to emphasize points based on their speech, and shifts posture in response to both the conversation's flow and predefined text instructions—all occurring in perfect harmony. This level of realism transforms the viewing experience, making interactions feel genuine and immersive. Achieving such lifelike behavior in virtual environments is no small feat, yet it is essential for enhancing user engagement in applications ranging from virtual reality to interactive gaming and beyond.

Driving a 3D avatar to perform such lifelike motions involves managing multiple control signals, such as text descriptions, speech audio, and trajectory data. Particularly, multi-modal signals may be provided concurrently, for instance, a text prompt like "a man is walking" alongside a speech audio clip. However, most prior works primarily focus on single-modality control, such as text-to-motion (Guo et al., 2022; Tevet et al., 2022) or speech-to-gesture (Ginosar et al., 2019b; Yi et al., 2023). Recent studies (Zhou & Wang, 2023; Zhou et al., 2023; Zhang et al., 2024) have explored designing unified models capable of addressing multiple modality control signals by leveraging datasets from different generation tasks. Nevertheless, these models typically process only one modality at a time, combining motions conditioned on different inputs in a limited and sequential manner when multiple control signals are present.

The primary challenge in achieving simultaneous multi-modal control of motion generation is the lack of aligned multi-modal data. Generating speech gestures that not only match the input speech

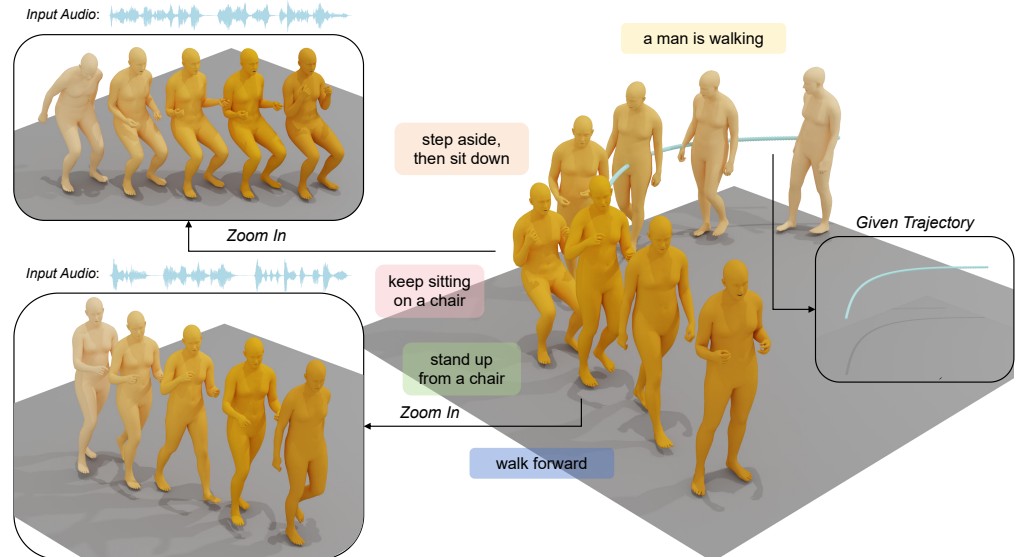

Figure 1: **Examples of Multi-Modal Controlled Motion Generation.** Given multiple control signals from different modalities—including text descriptions, speech audio, and trajectory data—our MOCO framework generates realistic and coherent holistic body motion. This includes both body movements and detailed features such as facial expressions and hand gestures, all closely aligned with the provided conditions. To clearly illustrate this, we highlight two clips with temporal zoom, showcasing the natural integration of speech gestures and lower-body movements in our generated motions.

audio but also align with the text description is difficult without such datasets. Additionally, the activity regions in speech-to-gesture datasets are often limited, making it hard to train models that can generate trajectory-controlled speech gestures. While collecting additional multi-modal data could help, it requires significant resources and remains constrained to specific scenarios. Some efforts, such as Yang et al. (2024), attempt to address this issue by combining the predictions of text-conditioned model and audio-conditioned model through weighted averaging, but this approach often results in mismatched and unrealistic motion sequences. Similarly, Ling et al. (2023) address this problem by generating pseudo text descriptions of a speaker's speech, including both the speaker's voice and spoken content (e.g., "A male speaker is saying: 'I am shocked by what you have done.'"), and replace scripts with movement descriptions during inference. However, the applicability of this method is strictly limited due to the constrained variety of pseudo labels.

To overcome these challenges, we propose a novel diffusion-based framework, **M**ulti-**MO**dal **C**ontrolled **CO**herent Motion Synthesis (MOCO). Inspired by Athanasiou et al. (2023) and Petrovich et al. (2024), our approach decouples the motion generation process during inference by independently modeling each modality. Specifically, speech audio naturally guides upper-body motion—like gestures and facial expressions—while text descriptions influence lower-body movements like walking or shifting stance. Our framework is first trained on multiple datasets, ensuring that the model can independently generate motions conditioned on either text or speech inputs. At each denoising step, the model generates motions for each modality separately from the input noise and assembles the body parts according to predefined spatial rules, i.e. combining audio-conditioned upper-body motion with text-conditioned lower-body motion to produce the combined motion. This combined motion is then diffused and used as the input noise for the next denoising step. The separation ensures that each body part's motion is highly aligned with its corresponding input condition, while the iterative process conditions each generation step on the current state of the combined motion. This allows each modality to refine its contribution within the context of the overall movement. Consequently, with each iteration, the motions generated for different body parts become increasingly harmonized, resulting in natural and fluid movements that exhibit coherent and synchronized behaviors. Furthermore, this decoupled generation process enables our framework to incorporate trajectory control into co-speech motion generation. We can leverage trajectory data to generate text-conditioned motion

and combine it with audio-conditioned motion, producing speech gestures that closely align with the given trajectory.

To the best of our knowledge, our method is one of the first to explicitly address the challenge of simultaneous multi-modal control in motion generation. To facilitate the evaluation of this novel task, we developed a multi-modal benchmark comprising 1,000 test clips which are generated from 40 fundamental text descriptions of body movements (e.g., "walk forwards" and "step back and sit down") and 694 audio clips from eight different speakers. Each test clip integrates two text prompts describing a movement with two speech audio clips. We rigorously evaluated our approach against baseline methods using both text-to-motion and speech-to-gesture metrics. Experimental results demonstrate that our method significantly outperforms existing baselines, advancing the field of multi-modal controlled motion generation for 3D avatars.

## 2 RELATED WORK

### 2.1 MULTI-MODAL CONDITIONED MOTION GENERATION

In recent years, human motion generation has received significant attention, driven in large part by advancements in dataset collection. Various scenarios have been explored depending on the input conditions, including action labels (Guo et al., 2020), text descriptions (Guo et al., 2022; Tevet et al., 2022; Zhang et al., 2022; Chen et al., 2023), speech audio (Ginosar et al., 2019b; Yi et al., 2023; Liu et al., 2023a; 2024), music (Li et al., 2021; Siyao et al., 2022; Tseng et al., 2023), scene context (Hassan et al., 2019; Ma et al., 2024), trajectory data (Xie et al., 2023), and even the motion of another person (Liu et al., 2023b). Beyond single-modality control, several works have aimed to handle multiple control signals. For example, Yoon et al. (2020) take into account speaker identity, speech audio, and transcripts to generate conversational gestures, while Yi et al. (2024) proposes a method for generating motion from both text and scene inputs. Moreover, some research has focused on integrating various datasets to train unified motion models that enhance scalability and applicability across multiple scenarios (Zhou & Wang, 2023; Zhang et al., 2024).

Despite these advancements, effectively managing concurrent multi-modal control signals remains challenging due to the scarcity of aligned multi-modal data. This limitation hampers the ability to generate coherent motions in scenarios that require the integration of multiple inputs, such as combining text descriptions with speech audio or integrating speech audio with trajectory data. To address this, Yang et al. (2024) propose combining predictions from text-conditioned and audio-conditioned models through weighted averaging. Similarly, Ling et al. (2023) suggest using speech scripts as pseudo text labels to create aligned text-audio-motion datasets, replacing scripts with movement descriptions during inference. However, these approaches are often constrained by biases in co-speech motion datasets, limiting their generalizability across diverse contexts.

### 2.2 DIFFUSION MODEL IN MOTION GENERATION

As one of the most advanced generative paradigms, diffusion models have gained significant traction in the field of human motion generation. Zhang et al. (2022) first introduced MotionDiffuse, a diffusion model that enables multi-level manipulation, including fine-grained control of body parts and arbitrary-length motion synthesis based on time-varying text prompts. More recently, Tevet et al. (2022) presented the Motion Diffusion Model (MDM), a transformer-based diffusion model featuring innovations such as predicting the sample itself rather than the noise, and incorporating geometric losses like foot contact loss to improve realism. Additionally, Chen et al. (2023) proposed a latent-based diffusion model, where the diffusion process operates in a learned latent space, enhancing the representation of motion. Following these foundational works, diffusion models have been applied across various motion generation scenarios, such as music-to-dance (Alexanderson et al., 2023), speech-to-gesture (Zhu et al., 2023), scene-conditioned motion generation (Huang et al., 2023), and human-human interaction (Liang et al., 2024).

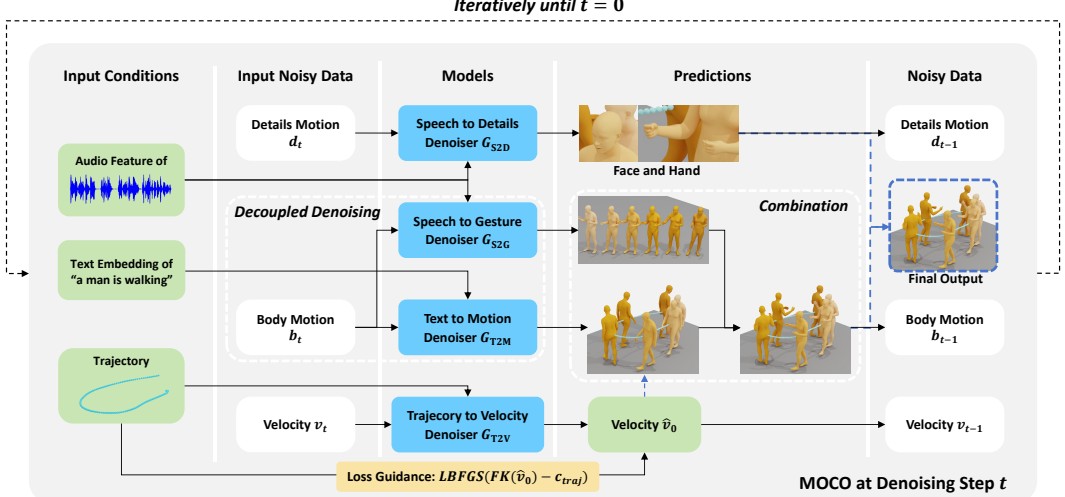

Figure 2: **Overview of MOCO.** At each denoising step $t$, input conditions and noisy data are fed into their respective denoisers to predict clean motion, which is then diffused for the next iteration. Specifically, the upper-body motion conditioned on speech audio and the lower-body motion conditioned on text description are combined to form the overall body motion. The blue arrows in the figure highlight two key points. One indicates that the denoising process of $v_0$ is completed before body motion denoising. The other shows that after the denoising process, the detailed facial and hand movements, and the combined body motion are integrated together to produce the final holistic motion.

## 3 METHOD

Given a set of condition signals and their corresponding time intervals, our framework generates realistic and coherent holistic body motions that precisely align with each condition within its specified time frame. To provide a comprehensive overview of our framework, we begin with a brief introduction to the Motion Diffusion Model (MDM) (Tevet et al., 2022), which serves as the foundational model in our approach (Section 3.1). Next, we describe the data representation and the various model modules employed in our framework (Section 3.2). Following this, we explain our multi-modal decoupled denoising for holistic body generation in scenarios where text and speech audio conditions are provided within the same time interval (Section 3.3). Finally, we address a more complex scenario where trajectory data is included, and each condition may have different time intervals (Section 3.4).

### 3.1 PRELIMINARY: MOTION DIFFUSION MODEL

Our work builds upon the denoising framework of the Motion Diffusion Model (MDM) (Tevet et al., 2022), which models diffusion as a Markov noising process $\{x_t\}_{t=0}^T$ starting from a sample $x_0$ from the data distribution. The transition between steps is defined by:

$$q\left(x_t \mid x_{t-1}\right) = \mathcal{N}\left(\sqrt{\alpha_t}\, x_{t-1},\ (1-\alpha_t)\mathbf{I}\right), \tag{1}$$

where $\alpha_t \in (0, 1)$, $\mathcal{N}(0, \mathbf{I})$ is a standard normal distribution, and $\mathbf{I}$ represents the identity matrix. As $t$ increases, the distribution of $x_T$ approaches $\mathcal{N}(0, \mathbf{I})$.

The primary objective of MDM is to model the conditional distribution $p(x_0 \mid c)$ by reversing this diffusion process through iterative denoising of $x_T$. To achieve this, MDM minimizes the following loss function:

$$\mathcal{L} = \mathbb{E}_{x_0, t}\left[\|x_0 - G\left(x_t, t, c\right)\|_2^2\right], \tag{2}$$

where $G$ is the denoiser. Sampling from $p(x_0 \mid c)$ is performed iteratively. At each timestep $t$, MDM predicts $x_0 = G(x_t, t, c)$ and computes $x_{t-1}$. This process continues until $t = 0$.

Additionally, MDM employs classifier-free guidance (Ho & Salimans, 2022) to control the influence of the conditioning signal $c$. The denoiser $G$ is trained on both conditioned and unconditioned data by randomly setting $c = \emptyset$ for a subset of training samples. This approach allows $G(x_t, t, \emptyset)$ to approximate the unconditional distribution. During sampling, MDM adjusts the strength of the conditioning signal using a scaling factor $s$ as follows:

$$G^s(x_t, t, c) = G(x_t, t, \emptyset) + s \cdot (G(x_t, t, c) - G(x_t, t, \emptyset)), \tag{3}$$

where $G^s$ denotes the sampling with classifier-free guidance for denoiser $G$. This technique enables precise control over how strongly the generated motion adheres to the conditioning signal, enhancing the model's ability to produce contextually appropriate motions.

## 3.2 DATA REPRESENTATION AND MODEL ARCHITECTURE

**Data Representation.** Our framework incorporates four main data modalities: motion, text, audio, and trajectory. The motion data is represented as $m = \{m^n\}|_{n=1}^N \in \mathcal{R}^{N \times 491}$, where $N$ is the number of frames. Specifically, the motion data for each frame is denoted as $m^n = \{b^n, d^n\}$, with $b \in \mathcal{R}^{205}$ representing the body pose (Petrovich et al., 2024), in which $v \in \mathcal{R}^3$ is the linear velocities of the pelvis in the $x$ and $y$ directions and the angular velocity around the body's vertical axis (Z-axis), and $d \in \mathcal{R}^{286}$ capturing detailed facial expression and hand movements. The text embeddings are encoded using a pretrained CLIP model (Radford et al., 2021) and are denoted as $c_{text} \in \mathcal{R}^{512}$. Audio features are extracted via a pretrained Wav2Vec2 model (Baevski et al., 2020) and represented as $c_{audio} \in \mathcal{R}^{N \times 768}$. Finally, the trajectory data is encoded as $c_{traj} \in \mathcal{R}^{N \times 2}$, representing the position on the XY-plane for each frame.

**Model Design.** Our framework includes four transformer-based denoisers: one for text-to-motion (T2M), one for speech-to-gesture (S2G), one for trajectory-to-velocity (T2V), and one for speech-to-details (S2D), which handles facial expressions and hand poses:

$$\hat{b}_0 = G_{\text{T2M}}(b_t, t, c_{text}) \tag{4}$$

$$\hat{b}_0 = G_{\text{S2G}}(b_t, t, c_{audio}) \tag{5}$$

$$\hat{v}_0 = G_{\text{T2V}}(v_t, t, c_{traj}) \tag{6}$$

$$\hat{d}_0 = G_{\text{S2D}}(d_t, t, c_{audio}). \tag{7}$$

We denote the sampling with classifier-free guidance for each denoiser as $G_u^s$, where $u \in \{\text{T2M, S2G, T2V, S2D}\}$.

For the T2M denoiser, which uses the text embedding $c_{text}$ as a condition, we follow prior work by treating $c_{text}$ as a token and applying self-attention to incorporate semantic information into the motion generation process. In contrast, the S2G, T2V, and S2D denoisers handle sequential data as conditions and utilize cross-attention to accurately model the relationships between the input sequences and the generated motion. Additionally, for the T2M and S2G denoisers, which are responsible for generating body poses, we initialize them with pretrained parameters from STMC (Petrovich et al., 2024) and fine-tune them on the HumanML3D and BEATX datasets. This initialization promotes faster convergence and reduces training time. All denoisers adhere to the objective function and diffusion paradigm described in Section 3.1.

## 3.3 MULTI-MODAL DECOUPLED DENOISING FOR SYNCHRONOUS CONDITIONS

In this section, we introduce our multi-modal decoupled denoising approach for generating holistic body motion in scenarios where text and speech audio conditions are provided synchronously—that is, within the same time interval—as shown in Figure 3 (a). Notably, we design different generation strategies for body motion and detailed movements, such as facial expressions and hand gestures, due to the lack of detailed motion data in the HumanML3D text-to-motion dataset.

**Multi-Modal Controlled Body Motion Generation.** Few works have explored using multi-modal control signals across datasets to generate motion. Yang et al. (2024) combine the predictions of the text-conditioned model and the audio-conditioned model through weighted averaging:

$$\hat{b}_0 = \gamma \cdot G_{\text{T2M}}(b_t, t, c_{\text{text}}) + (1 - \gamma) \cdot G_{\text{S2G}}(b_t, t, c_{\text{audio}}), \tag{8}$$

Figure 3: **Examples of synchronous and asynchronous conditions.** Synchronous conditions occur when all condition signals are provided within the same time interval. In contrast, asynchronous conditions involve multiple conditions, each corresponding to different time intervals.

where $\gamma$ is a parameter controlling the balance between the text-conditioned and speech-conditioned models. However, this method may lead to motions that do not closely match the input conditions. Further experimental analysis is presented in Appendix C.

Drawing inspiration from previous works (Athanasiou et al., 2023; Petrovich et al., 2024) that decompose complex text prompts into simpler components associated with specific body parts during inference, we propose to decouple the generation process for multi-modal control. Specifically, speech audio naturally guides upper-body gestures (including head and arm poses), while text descriptions influence lower-body movements (including spine and leg poses) like walking or shifting stance.

Based on this observation, we develop our multi-modal decoupled denoising method. At the beginning of each denoising step, the framework generates motions for each modality separately from the source noise. The upper-body motion conditioned on the speech audio and the lower-body motion conditioned on the text description are then combined to generate the overall motion. Finally, the overall motion is diffused and used as the input noise for the subsequent denoising step. The entire procedure can be formulated as follows:

$$\hat{b}_0 = I \odot G^s_{\text{T2M}}(b_t, t, c_{\text{text}}) + (1 - I) \odot G^s_{\text{S2G}}(b_t, t, c_{\text{audio}}), \tag{9}$$

$$b_{t-1} = \sqrt{\alpha_{t-1}}\,\hat{b}_0 + \sqrt{1 - \alpha_{t-1}}\,\epsilon, \tag{10}$$

where $I \in \mathbb{R}^{205}$ is the body mask for text-conditioned motion, a binary vector with entries set to 1 for the lower body and 0 for the upper body; $\odot$ denotes element-wise multiplication. The term $\alpha_t = \prod_{s=1}^{t}(1 - \beta_s)$ represents the cumulative product of $(1 - \beta_s)$ up to timestep $t$, and $\beta_t$ is the variance schedule controlling the amount of noise added at each timestep. The variable $\epsilon \sim \mathcal{N}(0, \mathbf{I})$ is Gaussian noise sampled from a standard normal distribution.

The decoupled denoising allows each body part's motion to be precisely guided by its corresponding input condition, ensuring high fidelity to the control signals. Moreover, by conditioning each generation step on the current combined motion, the model enables each modality to iteratively refine its contribution in the context of the overall movement. As the process progresses, the motions generated for different body parts become increasingly synchronized, resulting in natural and coherent full-body movements.

**Detailed Facial and Hand Movement Generation.** Since HumanML3D lacks this kind of data, we train a specialized model $G_{\text{S2D}}$ on BEATX to generate these elements from speech. When no speech is provided, the specialized model generates facial expressions and hand movements from unconditioned distributions:

$$\hat{d}_0 = \begin{cases} G^s_{\text{S2D}}(d_t, t, c), \text{if } c = c_{audio} \\ G_{\text{S2D}}(d_t, t, \emptyset), \text{if } c \neq c_{audio} \end{cases} \tag{11}$$

### 3.4 Trajectory Integration and Asynchronous Conditions

Having completed the multi-modal decoupled denoising for synchronous conditions, we now extend our MOCO framework to tackle more complex scenarios, such as incorporating trajectory control and managing asynchronous conditions.

**Trajectory Control.** Following the approach of Petrovich et al. (2024), we represent the global transition of body pose using the velocity vector $v = [\dot{r}_x, \dot{r}_y, \dot{\theta}]$, where $\dot{r}_x$ and $\dot{r}_y$ are the linear velocities of the pelvis in the $x$ and $y$ directions, respectively, and $\dot{\theta}$ is the angular velocity about

the body's vertical (Z) axis. Given the trajectory data $c_{\text{traj}}$, we first predict $\hat{v}_0$ using Equation 6. To enhance prediction accuracy, we incorporate loss guidance into our method. During each denoising step for predicting the velocity vector, we compute $\hat{v}_0$ using Equation 6 and apply loss guidance as follows:

$$L_{\text{guidance}} = FK(\hat{v}_0) - c_{\text{traj}}, \tag{12}$$

where $FK$ represents the differentiable Forward Kinematics function that converts linear and angular velocities into the trajectory. We optimize $L_{\text{guidance}}$ with respect to $\hat{v}_0$ using the second-order LBFGS optimizer (Liu & Nocedal, 1989), following the methodology of Wang et al. (2023). This optimization ensures that the predicted global transitions closely match the provided trajectory data.

Once $\hat{v}_0$ is predicted based on $c_{\text{traj}}$, we substitute the velocity component in $\hat{b}_0$ with $\hat{v}_0$ during each iteration of its generation. This substitution guides the generation process to adapt the remaining elements of $\hat{b}_0$ to align with $\hat{v}_0$, thereby ensuring consistency with the provided trajectory data.

**Managing Asynchronous Conditions Timeline.** To extend our framework to broader applications where multiple conditions are provided and each corresponding to different time intervals, i.e. asynchronous conditions, we adopt a timeline-based strategy as described in Petrovich et al. (2024). Specifically, given a set of conditions and their corresponding time intervals, we denote them as $\{c_j, f_j^s, f_j^e\}$ for $1 \leq j \leq J$, where $c_j$ represents the $j$-th condition, and $f_j^s$ and $f_j^e$ are the respective start and end frames within the overall timeline. Here, $J$ is the total number of conditions.

During each denoising step $t$, the body pose over the entire timeline is generated as follows:

$$\hat{b}_0 = \sum_{j=1}^{J} I_j \odot G_j^s \left( b_{t, f_j^s : f_j^e}, t, c_j \right), \tag{13}$$

where $I_j$ is a binary mask corresponding to the motion generated by the $j$-th condition, and $G_j^s \in \{G_{\text{T2M}}^s, G_{\text{S2G}}^s\}$ represents the denoiser used for the $j$-th condition. The operator $\odot$ denotes element-wise multiplication.

Similarly, the denoising step $t$ for generating facial and hand movements across the entire timeline is expressed as:

$$\hat{d}_0 = \sum_{j=1}^{J} G_{\text{S2D}} \left( d_{t, f_j^s : f_j^e}, t, c_j \right). \tag{14}$$

In particular, if $c_j$ is a text condition, it is replaced with an unconditional condition $\emptyset$. The overall holistic body motion is then represented as $\hat{m}_0 = \{\hat{b}_0, \hat{d}_0\}$. This strategy enables our framework to handle multiple conditions over different time intervals, facilitating more flexible and complex motion generation scenarios. We further explore methods to generate smoother transitions at interval boundaries in Section 4.

## 4 EXPERIMENTS

### 4.1 DATASETS

**Task-Specific Datasets.** HumanML3D dataset is a large **Text-to-Motion** dataset created by amalgamating motion sequences from the HumanAct12 and AMASS datasets (Guo et al., 2022). It consists of 14,616 motions and 44,970 descriptions composed of 5,371 distinct words, totaling 28.59 hours of motion data. To align the data representation—specifically, to use SMPL-X parameters for representing joint rotations—we utilize only the AMASS portion of HumanML3D because it has an official SMPL-X version. BEATX dataset is a large-scale **Speech-to-Gesture** dataset specifically designed for research in speech-to-gesture generation (Liu et al., 2023a). It contains synchronized recordings of speech audio and corresponding 3D motion capture data of human gestures. In addition to audio and motion data, the dataset includes annotations such as text transcriptions and emotional states.

**Multi-Modal Benchmark.** To effectively evaluate our proposed task, we created a multi-modal benchmark consisting of 1,000 test clips by following the procedure outlined in Petrovich et al.

| | Text2Motion | | | | | Speech2Gesture | | | Transition |
|---|---|---|---|---|---|---|---|---|---|
| | FID+ ↓ | R1 ↑ | R3 ↑ | M2T ↑ | M2M ↑ | FID-A ↓ | BC ↑ | L1div ↑ | MTD ↓ |
| GT (Ground Truth) | 0.000 | 40.0 | 72.5 | 0.781 | 1.000 | - | - | - | 2.9 |
| Audio-Only | 1.647 | 2.9 | 8.6 | 0.514 | 0.507 | 2.19 | 2.45 | 4.51 | 1.2 |
| Text-Only | 0.587 | 27.1 | 53.3 | 0.730 | 0.702 | 5.30 | 1.90 | 6.78 | 4.9 |
| Weighted Average (Yang et al., 2024) | 1.335 | 6.8 | 16.1 | 0.546 | 0.537 | 2.17 | 2.20 | 4.08 | 1.2 |
| Pseudo-Text (Ling et al., 2023) | 1.593 | 2.2 | 7.0 | 0.511 | 0.503 | 2.22 | 2.55 | 6.43 | 1.7 |
| **MOCO** | 0.862 | 24.6 | 46.9 | 0.649 | 0.639 | 3.83 | 2.72 | 8.62 | 5.3 |

Table 1: Comparison with baselines.

(2024). Each test clip is automatically constructed and contains two text descriptions and two audio clips. To create these clips, we first manually collected a set of 40 texts focusing on lower-body movements that commonly occur during speech delivery or conversation . We then split the audio from the BEATX test set into clips using a Voice Activity Detector (VAD). To serve as ground truth for computing evaluation metrics (Section 4.2), we selected motion samples from AMASS and BEATX that correspond to each text and audio clip. Based on these atomic texts and audio clips, we automatically generated test clips.

## 4.2 METRICS

We evaluate our method using three categories: text-to-motion, speech-to-gesture, and transition smoothness (Liu et al., 2023a; Petrovich et al., 2024). For text-to-motion, *FID+* assesses realism by measuring the distribution difference between real and generated motions using five random 5-second clips per test sample. *R1* and *R3* metrics evaluate alignment by recording the frequency of correct text prompts appearing in the top-1 and top-3 retrieved texts, respectively. *M2T* (motion-to-text) and *M2M* (motion-to-motion) measure alignment through cosine similarity between embeddings of generated motions and ground truth texts or motions. In the speech-to-gesture category, *FID-A* similarly measures the realism of motion generated based on speech audio. *Beat Consistency (BC)* evaluates how well gestures synchronize with the rhythm and beats of the speech, while *L1 Diversity (L1Div)* quantifies gesture diversity by calculating the average L1 distance between multiple gesture clips. Transition smoothness is assessed by *Max Transition Distance (MTD)*, which measures the maximum distance between consecutive frames during transitions, with lower values indicating smoother and more realistic motions. This comprehensive set of metrics ensures a thorough evaluation of our method across key dimensions.

## 4.3 COMPARISON WITH BASELINES

In Table 1, we compare our proposed MOCO with several baseline methods, including *Audio-Only*, an audio-conditioned model trained exclusively on the speech-to-gesture dataset; *Text-Only*, a text-conditioned model trained solely on the text-to-motion dataset; *Weighted Average*, a method that follows Yang et al. (2024) by combining the predictions of text- and audio-conditioned models through weighted averaging; and *Pseudo-Text*, a method that follows Ling et al. (2023) by using pseudo text descriptions of a speaker's speech as the text condition during training.

As shown in the table, the single-modality baselines achieve the highest performance within their respective domains but perform poorly on the other modality's metrics. Specifically, the *Audio-Only* excels in speech-to-gesture metrics but underperforms in text-to-motion metrics, while the *Text-Only* performs well in text-to-motion metrics but poorly in speech-to-gesture metrics. In contrast, our proposed MOCO exhibits robust performance across both sets of metrics, delivering competitive results in both text-to-motion and speech-to-gesture tasks simultaneously. This underscores the effectiveness of MOCO in generating condition-aligned motions when multi-modal conditions are provided concurrently.

It is important to note that the Fréchet Inception Distance (FID) is computed based on the similarity between the generated data and the ground truth. For instance, MOCO's upper-body motion, which primarily consists of speech gestures, differs significantly from the ground truth in the text-to-motion dataset. Therefore, even though MOCO's generated lower-body motion closely follows the text descriptions (e.g., walking, standing, or sitting) similar to the *Text-Only*, the discrepancy in upper-body motion results in a higher FID+ compared to the *Text-Only*. Similarly, while MOCO's upper-body

| Method | Share Weight | Body Mask $1-I$ | Transition Method | Text2Motion | | | | | Speech2Gesture | | | Transition |
|---|---|---|---|---|---|---|---|---|---|---|---|---|
| | | | | FID+ ↓ | R1 ↑ | R3 ↑ | M2T ↑ | M2M ↑ | FID-A ↓ | BC ↑ | L1div ↑ | MTD ↓ |
| GT | - | - | - | 0.000 | 40.0 | 72.5 | 0.781 | 1.000 | - | - | - | 2.9 |
| **MOCO** | ✗ | head, arms | diffcollage | 0.862 | 24.6 | 46.9 | 0.649 | 0.639 | 3.83 | 2.72 | 8.62 | 5.3 |
| Variant 1 | ✗ | head, arms, spine | diffcollage | 0.921 | 22.4 | 44.2 | 0.634 | 0.617 | 3.86 | 2.81 | 8.35 | 4.7 |
| Variant 2 | ✗ | spine, legs | diffcollage | 1.234 | 7.9 | 18.7 | 0.554 | 0.550 | 2.75 | 2.19 | 5.11 | 1.6 |
| Variant 3 | ✓ | head, arms | diffcollage | 0.866 | 22.1 | 47.8 | 0.656 | 0.641 | 4.24 | 2.68 | 8.95 | 4.5 |
| Variant 4 | ✗ | head, arms | average | 0.858 | 24.0 | 46.5 | 0.650 | 0.639 | 3.83 | 2.68 | 8.56 | 6.5 |

Table 2: Ablation study on key designs within MOCO.

motion aligns well with the speech audio, as seen in the *Audio-Only*, differences in lower-body motion cause MOCO's FID-A to be larger than that of the *Audio-Only*.

The other two baselines, *Weighted Average* and *Pseudo-Text*, perform similarly to the *Audio-Only*, achieving good results on speech-to-gesture metrics but poor performance on text-to-motion metrics, indicating their limited ability to handle multi-modal data effectively. We explain this further in Appendix C.

## 4.4 ABLATION STUDY

To assess the impact of key designs within our MOCO framework, we conduct an ablation study presented in Table 2. This study systematically examines the effects of body masking (*Body Mask*), weight sharing (*Share Weight*), and transition methods (*Transition Method*) on the model's performance across text-to-motion and speech-to-gesture metrics.

**Body Masking.** In Variants 1 and 2, we test our hypothesis that speech audio guides upper-body motion (head and arms) while text descriptions influence lower-body movements (spine and legs). In Variant 1, we expand the body mask to include the spine along with the head and arms (*Body Mask* = head, arms, spine). This modification results in an increased FID+ and a slight decrease in R1 and R3, indicating a decline in text-to-motion performance. Moreover, it does not produce significant improvements in speech-to-gesture metrics, suggesting that including the spine in the body mask fails to enhance gesture generation and instead compromises text-driven motion performance.

Variant 2 further adjusts the body mask to include the legs and spine (*Body Mask* = legs, spine), leading to a significant deterioration in text-to-motion metrics and Beat Consistency. This decline primarily arises because the text descriptions in our multi-modal benchmark include various movements such as "walk," "sit," and "turn right," while the speech-to-gesture data predominantly features standing gestures, creating a substantial mismatch. Controlling lower-body motion with audio makes it difficult to align the motion with text descriptions, while controlling upper-body motion with text complicates alignment with beats. Although Variant 2 shows a notable improvement in FID-A, suggesting a bias in the speech-to-gesture data where most motions involve standing in place, the overall performance deteriorates.

In contrast, our original method (*Body Mask* = head, arms) effectively balances the influences of both text and audio inputs. By assigning the upper body to be guided by audio and the lower body by text, we achieve superior results across both text-to-motion and speech-to-gesture metrics. This demonstrates the advantage of our approach in producing coherent and contextually appropriate motions that align well with the provided conditions.

**Weight Sharing.** In Variant 3, we enable weight sharing (*Share Weight* = ✓), following previous multi-modal methods (Ling et al., 2023; Yang et al., 2024), while keeping the body mask and transition method unchanged. Compared to the full MOCO model (without weight sharing), enabling weight sharing results in poorer performance across several metrics, including R1 and FID-A. This decline suggests that sharing weights between modalities may limit the model's ability to capture modality-specific nuances, thereby reducing its effectiveness in generating accurate and realistic motions for both text-to-motion and speech-to-gesture tasks.

**Transition Methods.** For ensuring smooth transitions between motion segments, we adopt "diffcollage" (Zhang et al., 2023c), as utilized by Petrovich et al. (2024). This method creates an overlap area at the transition point and combines conditional and unconditional predictions within this region to

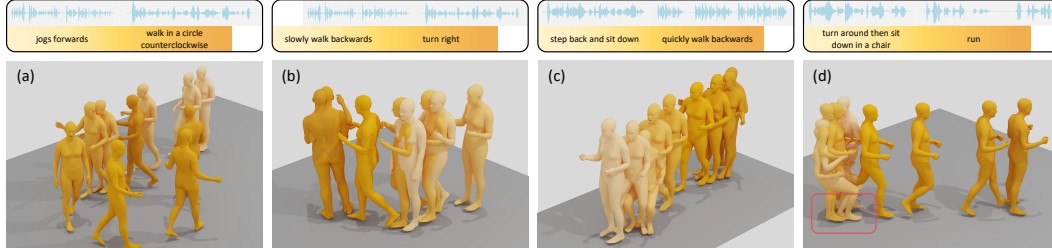

Figure 4: **Qualitative results.** We visualize four samples generated by MOCO. Darker colors represent later points in time. The results demonstrate that MOCO is capable of generating coherent and realistic motions that highly align with the given multi-modal control signals. Figures (a), (b), and (d) present natural speech gestures coordinated with various lower-body movements as specified by the text inputs, such as jogging, walking in a circle, turning right, running, and so on. Figure (c) displays natural movements of delivering a speech while sitting down. Figure (d) reveals a limitation of MOCO. When standing up or sitting down, the foot should remain stationary. However, the foot highlighted in the red box slides, leading to unrealistic results. This issue should be addressed in future work.

achieve seamless motion continuity. We compare diffcollage with an alternative transition method in the Variant 4: average", which applies a weighted average in the overlap area. This approach results in a slight improvement in FID+ compared to the full MOCO text2motion modelut leads to a decrease in transition smoothness, indicating more abrupt transitions between motion intervals. This suggests that while the edit method may marginally enhance certain performance metrics, it compromises the fluidity of motion, which is crucial for realistic motion synthesis.

### 4.5 QUALITATIVE ANALYSIS

To clearly illustrate the overall performance of MOCO, we visualize four samples generated by MOCO along with their corresponding conditions in Figure 4. The lighter color of the mesh and the background of the text description indicate the start of the sequence, while the darker color indicates the end of the sequence. These results showcase natural speech upper-body gestures that coordinate with various lower-body motions such as jogging, walking, and sitting, indicating that MOCO is capable of generating coherent and realistic motions that highly align with the given multi-modal control signals. Please see the caption for a full analysis of these examples.

## 5 CONCLUSION

In this study, we present **MOCO**, a novel diffusion-based framework to generate realistic and coherent holistic body motions from multi-modal inputs, including text descriptions, speech audio, and trajectory data. Our key innovation lies in a decoupled denoising process where, during each denoising step, the model independently generates motions for each modality and assembles them according to predefined spatial rules. This approach ensures that the generated motion is closely aligned with each condition while producing realistic and coherent whole-body movements. Experimental results demonstrate that our approach delivers state-of-the-art performance both qualitatively and quantitatively, advancing the field of multi-modal controlled motion generation for 3D avatars.

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

## A  THEORETICAL ANALYSIS FOR DECOUPLE-THEN-COMBINE

Our proposed MOCO relies on the assumption that the joint conditional probability $p(x_{t-1} \mid c_{\text{text}}, c_{\text{audio}}, x_t)$ can be approximated by $p(x_{t-1,\text{lower}} \mid c_{\text{text}}, x_t) \cdot p(x_{t-1,\text{upper}} \mid c_{\text{audio}}, x_t)$, expressed as:

$$p(x_{t-1} \mid c_{\text{text}}, c_{\text{audio}}, x_t) \approx p(x_{t-1,\text{lower}} \mid c_{\text{text}}, x_t) \cdot p(x_{t-1,\text{upper}} \mid c_{\text{audio}}, x_t), \tag{15}$$

where $x_t$ denotes the motion at denoising step $t$, composed of upper-body motion $x_{t,\text{upper}}$ and lower-body motion $x_{t,\text{lower}}$.

We provide a detailed derivation of Equation 15, outlining the two approximations involved in the decomposition process. The derivation follows these steps:

$$
\begin{aligned}
p(x_{t-1} \mid c_{\text{text}}, c_{\text{audio}}, x_t) &= p(x_{t-1,\text{lower}}, x_{t-1,\text{upper}} \mid c_{\text{all}}), \text{ where } c_{\text{all}} = \{c_{\text{text}}, c_{\text{audio}}, x_t\} \\
&= p(x_{t-1,\text{lower}} \mid c_{\text{all}}) \cdot p(x_{t-1,\text{upper}} \mid c_{\text{all}}, x_{t-1,\text{lower}}) \tag{16} \\
&\approx p(x_{t-1,\text{lower}} \mid c_{\text{all}}) \cdot p(x_{t-1,\text{upper}} \mid c_{\text{all}}) \tag{17} \\
&\approx p(x_{t-1,\text{lower}} \mid c_{\text{all}} \setminus \{c_{\text{audio}}\}) \cdot p(x_{t-1,\text{upper}} \mid c_{\text{all}} \setminus \{c_{\text{text}}\}) \tag{18} \\
&= p(x_{t-1,\text{lower}} \mid c_{\text{text}}, x_t) \cdot p(x_{t-1,\text{upper}} \mid c_{\text{audio}}, x_t).
\end{aligned}
$$

The first approximation occurs in the transition from Equation 16 to Equation 17. Here, we approximate:

$$
\begin{aligned}
p(x_{t-1,\text{upper}} \mid c_{\text{all}}, x_{t-1,\text{lower}}) &= p(x_{t-1,\text{upper}} \mid c_{\text{text}}, c_{\text{audio}}, x_{t,\text{upper}}, x_{t,\text{lower}}, x_{t-1,\text{lower}}) \\
&\approx p(x_{t-1,\text{upper}} \mid c_{\text{text}}, c_{\text{audio}}, x_{t,\text{upper}}, x_{t,\text{lower}}) \\
&= p(x_{t-1,\text{upper}} \mid c_{\text{all}}).
\end{aligned}
$$

This approximation assumes that $x_t$ already encapsulates sufficient information about $x_{t-1}$, allowing us to neglect the influence of $x_{t-1,\text{lower}}$ when estimating $x_{t-1,\text{upper}}$. This simplification is justified by the proximity of the diffusion steps and the strong correlation between the states at steps $t$ and $t-1$.

The second approximation occurs in the transition from Equation 17 to Equation 18, where we decouple modality-specific influences:

$$
\begin{aligned}
p(x_{t-1,\text{lower}} \mid c_{\text{all}}) &\approx p(x_{t-1,\text{lower}} \mid c_{\text{all}} \setminus \{c_{\text{audio}}\}), \\
p(x_{t-1,\text{upper}} \mid c_{\text{all}}) &\approx p(x_{t-1,\text{upper}} \mid c_{\text{all}} \setminus \{c_{\text{text}}\}).
\end{aligned}
$$

This approximation leverages the observation that text input ($c_{\text{text}}$) primarily influences lower-body movements (e.g., walking or shifting stance), while audio input ($c_{\text{audio}}$) predominantly affects upper-body movements (e.g., gestures or facial expressions). By excluding $c_{\text{audio}}$ from the conditioning set for $x_{t-1,\text{lower}}$ and $c_{\text{text}}$ for $x_{t-1,\text{upper}}$, we ensure the conditioning focuses on the most relevant modality for each body part.

## B  RULES FOR MANAGING MULTI-MODAL ASYNCHRONOUS CONDITIONS

In this section, we outline the rules of the MOCO framework for managing multi-modal asynchronous conditions. Our rules build upon the excellent work of STMC (Petrovich et al., 2024) and extend them to accommodate multi-modal scenarios.

**Default**:

1. **Single Active Condition**: When only one condition is active, it governs the movement of the entire body.
2. **Two Active Conditions of Different Modalities**: When two conditions from different modalities (e.g., speech and text) are active simultaneously, speech by default controls upper body movements (i.e., head and arms), while text by default controls lower body movements (i.e., legs and spine).

**Flexible**:

To achieve more nuanced control, we leverage STMC's rules. When two conditions are active simultaneously:

1. **Different Body Parts**: If the conditions control different body parts, each condition governs its respective parts without conflict.

2. **Overlapping Body Parts**: If both conditions attempt to control the same body parts, the condition controlling fewer body parts takes precedence for those specific parts.

3. **Equal Control Scope**: If both conditions control an equal number of body parts, the condition with the earlier start time takes precedence. The later-starting condition will only control movement after the earlier condition has concluded.

## C LIMITATIONS OF WEIGHTED AVERAGING IN MULTI-MODAL MOTION GENERATION

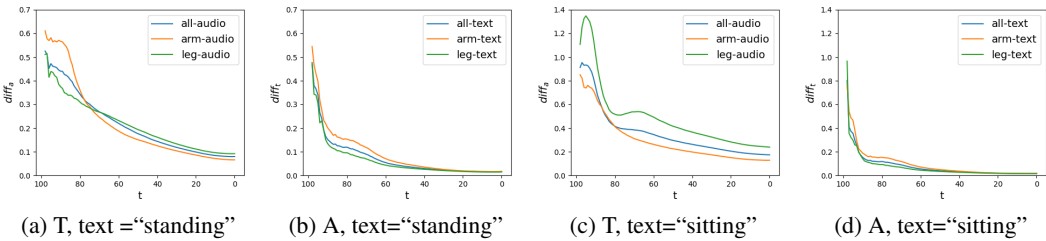

(a) T, text ="standing"  (b) A, text="standing"  (c) T, text="sitting"  (d) A, text="sitting"

Figure 5: Comparison of differences calculated by the speech-to-gesture model and the text-to-motion model during motion updates. "**T**" denotes using text-to-motion model to update motion, while "**A**" denotes using speech-to-gesture to update motion. The results show that the speech-to-gesture model computes larger differences than the text-to-motion model, indicating it adjusts the motion more aggressively based on the conditions. This explains why the weighted averaging method's generated results closely resemble those produced entirely by the speech-to-gesture model. Additionally, when the text condition is "sitting," the speech-to-gesture model calculates larger differences in the legs than in the arms, which is counterintuitive and may be attributed to data bias in the speech-to-motion dataset.

To understand why the *Weighted Average* method perform similarly to the *Audio-Only*—achieving good results in speech-to-gesture metrics but poor performance in text-to-motion metrics—we conducted the following experiments.

Given both speech and text inputs, we updated the motion using only the text-to-motion model. At each denoising step $t$, we computed the difference $\text{diff}_t$ between the speech-to-gesture model's prediction—based on the speech input and the current motion from the text-to-motion model—and the current motion from the text-to-motion model. This difference quantifies how much the speech-to-gesture model perceives a mismatch between the speech condition and the current motion. Conversely, when we used only the speech-to-gesture model to update the motion, the calculated difference indicated how much the text-to-motion model perceived a mismatch between the text condition and the current motion. A larger difference suggests a greater mismatch and that the model will update the motion more aggressively.

We recorded these differences in both scenarios and divided them into whole body, arms, and legs for clearer illustration. Comparing Figures 5 (a) and (b), as well as Figures 5 (c) and (d), we found that the differences calculated by the speech-to-gesture model are larger than those by the text-to-motion model. This indicates that the speech-to-gesture model adjusts the motion more aggressively based on its conditions than the text-to-motion model does. This explains why, when using the weighted averaging method, the generated result closely resembles that produced entirely by the speech-to-gesture model.

Furthermore, by comparing Figures 5 (a) and (c), which have different text conditions, we observe that when the text condition is "sitting," the differences calculated by the speech-to-gesture model in the legs are larger than in the arms. This is counterintuitive since speech is typically associated with upper-body gestures rather than lower-body movements. Conversely, when the text condition is "standing," the differences in the legs are smaller than in the arms, aligning with expectations. This

phenomenon may be attributed to data bias in the speech-to-motion dataset, where most motions are performed in standing positions.

These observations reveal the limitations of weighted averaging in multi-modal motion generation and suggest the validity of our proposed decoupled denoising process.

# D    COMPUTATIONAL COMPLEXITY

|  | Parameters (M) | Model Size (MB) | FLOPs (G) | Inference Time (ms/frame) |
|---|---|---|---|---|
| $G_{\text{T2M}}$ | 27.01 | 103.02 | 5.19 | 2.26 |
| $G_{\text{S2G}}$ | 36.86 | 140.62 | 6.72 | 4.30 |
| $G_{\text{T2V}}$ | 0.34 | 1.31 | 0.06 | 6.20 |
| $G_{\text{S2D}}$ | 36.94 | 140.94 | 6.74 | 4.37 |

Table 3: Complexity of each denoiser of MOCO.

Our framework, MOCO, comprises four transformer-based denoisers: $G_{\text{T2M}}$ for text-to-motion (T2M), $G_{\text{S2G}}$ for speech-to-gesture (S2G), $G_{\text{T2V}}$ for trajectory-to-velocity (T2V), and $G_{\text{S2D}}$ for speech-to-details (S2D), which manages facial expressions and hand poses. To clearly illustrate the computational complexity of MOCO, we present various metrics, including the number of parameters, model size, FLOPs, and inference time on a single NVIDIA 4090 GPU, as shown in Table 3.

As indicated in the table, our framework is overall lightweight and sufficiently fast. Specifically, the speech-to-gesture denoiser $G_{\text{S2G}}$ and the speech-to-details denoiser $G_{\text{S2D}}$ are relatively larger than the other denoisers due to additional cross-attention parameters. In contrast, the trajectory-to-velocity denoiser $G_{\text{T2V}}$ is the most lightweight module, featuring fewer hidden state dimensions and transformer layers because the task it handles involves low-dimensional data. However, the introduction of a guidance mechanism for more accurate predictions results in $G_{\text{T2V}}$ having the longest inference time.

Finally, to generate the motion sequences for a 35-second demo video consisting of nine clips under different conditions and with a total duration of 54 seconds, our method completed the body motion generation task in only 3.72 seconds. This fast generation time highlights the potential of our approach for real-time applications.

# E    ADDITIONAL EXPERIMENTS

## E.1    EVALUATION OF TRAJECTORY CONTROL

| Method | | Location | | Orientation | |
|---|---|---|---|---|---|
| CFG | L-BFGS | Average Difference | Goal Difference | Average Difference | Goal Difference |
| ✗ | ✗ | 0.5641 | 1.2068 | 0.7059 | 1.2583 |
| ✓ | ✗ | 0.5676 | 1.3177 | 0.8276 | 1.5115 |
| ✗ | ✓ | **0.0747** | **0.1235** | **0.6009** | **1.1031** |
| ✓ | ✓ | 0.1121 | 0.1950 | 0.7111 | 1.2845 |

Table 4: Evaluation of Trajectory Control.

Table 4 evaluates trajectory control methodologies by assessing the effects of classifier-free guidance (CFG) and L-BFGS optimization on both location (meters) and orientation (radians). For each category, two primary metrics are reported: Average Difference, quantifying the mean deviation between the generated trajectory and the ground truth (GT), and Goal Difference, measuring the discrepancy at the final point relative to the GT. The results show that L-BFGS optimization significantly reduces location differences and modestly improves orientation accuracy. Notably, for the same trajectory, orientation can be diverse, so the generated orientation does not need to closely match the GT. In contrast, incorporating CFG does not enhance trajectory accuracy. These findings indicate that while L-BFGS is a robust optimization strategy for trajectory control, integrating CFG may not provide complementary advantages and could interfere with the optimization process.

| | Text2Motion | | | | | Speech2Gesture | | | Transition |
|---|---|---|---|---|---|---|---|---|---|
| | FID+↓ | R1↑ | R3↑ | M2T↑ | M2M↑ | FID-A↓ | BC↑ | L1div↑ | MTD↓ |
| Ground Truth | 0.000 | 40.0 | 72.5 | 0.781 | 1.000 | - | - | - | 2.9 |
| Synchronous | 0.896 | 23.8 | 45.9 | 0.638 | 0.629 | 4.41 | 2.62 | 9.47 | 5.5 |
| Asynchronous | 0.862 | 24.6 | 46.9 | 0.649 | 0.639 | 3.83 | 2.72 | 8.62 | 5.3 |

Table 5: Comparison of MOCO in synchronous and asynchronous conditions.

## E.2 Performance under Synchronous and Asynchronous Conditions

In Table 5, we compare the performance of MOCO under synchronous and asynchronous conditions. As illustrated in the table, MOCO generates slightly better motions under asynchronous conditions compared to synchronous ones. This improvement may be attributed to asynchronous conditions allowing a single modality to control the entire body, rather than using multiple modalities to control different parts simultaneously. Such an approach is likely simpler for the model, as it was trained on data where single modalities govern the whole body. Additionally, motions generated under single-modality conditions more closely align with the distribution of the GT in the test set, which also consists of motions under single-modality conditions. Consequently, this alignment results in better performance metrics.

## E.3 Single Modality Performance

| Methods | R-Precision | | | FID↓ | MM Dist↓ | Diversity↑ | MM↑ |
|---|---|---|---|---|---|---|---|
| | Top 1 | Top 2 | Top 3 | | | | |
| Ground Truth | $0.511^{\pm.003}$ | $0.703^{\pm.003}$ | $0.797^{\pm.002}$ | $0.002^{\pm.000}$ | $2.974^{\pm.008}$ | $9.503^{\pm.065}$ | - |
| T2M-GPT (Zhang et al., 2023a) | $0.491^{\pm.003}$ | $0.680^{\pm.003}$ | $0.775^{\pm.002}$ | $0.116^{\pm.004}$ | $3.118^{\pm.011}$ | $9.761^{\pm.081}$ | $1.856^{\pm.011}$ |
| MDM (Tevet et al., 2022) | - | - | $0.611^{\pm.007}$ | $0.544^{\pm.044}$ | $5.566^{\pm.027}$ | $9.559^{\pm.086}$ | $2.799^{\pm.072}$ |
| **MOCO (Ours)** | $0.434^{\pm.010}$ | $0.618^{\pm.008}$ | $0.720^{\pm.008}$ | $0.530^{\pm.044}$ | $3.563^{\pm.049}$ | $9.856^{\pm.166}$ | $2.663^{\pm.068}$ |
| FineMoGen (Zhang et al., 2023b) | $0.504^{\pm.002}$ | $0.690^{\pm.002}$ | $0.784^{\pm.002}$ | $0.151^{\pm.008}$ | $2.998^{\pm.008}$ | $9.263^{\pm.094}$ | $2.696^{\pm.079}$ |
| MoMask (Guo et al., 2024) | $0.521^{\pm.002}$ | $0.713^{\pm.002}$ | $0.807^{\pm.002}$ | $0.045^{\pm.002}$ | $2.958^{\pm.002}$ | - | $1.241^{\pm.040}$ |
| LMM-Tiny (Zhang et al., 2025) | $0.496^{\pm.002}$ | $0.685^{\pm.002}$ | $0.785^{\pm.002}$ | $0.415^{\pm.002}$ | $3.087^{\pm.012}$ | $9.176^{\pm.074}$ | $1.465^{\pm.048}$ |
| LMM-Large (Zhang et al., 2025) | $0.525^{\pm.002}$ | $0.719^{\pm.002}$ | $0.811^{\pm.002}$ | $0.040^{\pm.002}$ | $2.943^{\pm.012}$ | $9.814^{\pm.076}$ | $2.683^{\pm.054}$ |

Table 6: Quantitative results of text-to-motion generation on the HumanML3D test set.

| Methods | FGD↓ | BC | Diversity↑ | MSE↓ | LVD↓ |
|---|---|---|---|---|---|
| FaceFormer (Fan et al., 2022) | - | - | - | 7.787 | 7.593 |
| CodeTalker (Xing et al., 2023) | - | - | - | 8.026 | 7.766 |
| S2G (Ginosar et al., 2019a) | 28.15 | 4.683 | 5.971 | - | - |
| Trimodal (Yoon et al., 2020) | 12.41 | 5.933 | 7.724 | - | - |
| HA2G (Liu et al., 2022c) | 12.32 | 6.779 | 8.626 | - | - |
| DisCo (Liu et al., 2022a) | 9.417 | 6.439 | 9.912 | - | - |
| CaMN (Liu et al., 2022b) | 6.644 | 6.769 | 10.86 | - | - |
| DiffStyleGesture (Yang et al., 2023) | 8.811 | 7.241 | 11.49 | - | - |
| TalkShow (Yi et al., 2023) | 6.209 | 6.947 | 13.47 | 7.791 | 7.771 |
| EMAGE (Liu et al., 2023a) | 5.512 | 7.724 | 13.06 | 7.680 | 7.556 |
| ProbTalk (Liu et al., 2024) | 6.170 | 8.099 | 10.43 | 8.990 | 8.385 |
| MOCO (Ours) | 5.543 | 7.089 | 14.05 | 7.285 | 7.573 |

Table 7: Quantitative results of speech-to-gesture generation on the BEATX test set.

To demonstrate MOCO's performance in single-modality scenarios, we trained it from scratch on HumanML3D for text-to-motion and on BEATX for speech-to-gesture, respectively, ensuring a fair comparison. The results, presented in Tables 6 and 7, show that in the HumanML3D text-to-motion benchmark (Table 6), our model achieves performance comparable to the widely-used MDM. This outcome is expected since our text-to-motion denoiser, $G_{\text{T2M}}$, is based on MDM. In the BEATX speech-to-gesture benchmark (Table 7), MOCO attains competitive performance compared to state-of-the-art methods.

| Method | Better Text Following (%) | Better Beat Synchronization (%) |
|---|---|---|
| Neither | 1.0 | 13.0 |
| Pseudo-Text | 0.0 | 12.5 |
| MOCO (Ours) | 99.0 | 74.5 |
| Neither | 0.0 | 13.5 |
| Weighted Average | 0.0 | 3.5 |
| MOCO (Ours) | 100.0 | 83.0 |
| | Better Body Coherence (%) | Better Temporal Fluidity (%) |
| Neither | 12.0 | 12.6 |
| Combine Only Last Time | 17.0 | 42.6 |
| MOCO (Ours) | 71.0 | 44.8 |

Table 8: User study.

### E.4 USER STUDY

Table 8 presents the results of a user study comparing MOCO with three baseline methods. Specifically, we evaluate MOCO against *Pseudo-Text* and *Weighted Average*, introduced in Section 4.3, to assess overall performance in text following and audio beat synchronization. Additionally, we compare MOCO with *Combine Only Last Time*, which also employs the decoupling strategy but applies the combining strategy only at the final diffusion step. This comparison aims to evaluate whole body coherence and temporal fluidity.

As shown in the table, MOCO achieves significant advantages over both *Pseudo-Text* and *Weighted Average*, demonstrating the effectiveness of our decouple-then-combine strategy in generating motion aligned with multi-modal conditions. Furthermore, when compared to *Combine Only Last Time*, our method was rated significantly higher in both body coherence and temporal fluidity. This indicates that MOCO does more than merely combine different body parts controlled by separate conditions; it ensures that each body part aligns with its corresponding condition while enhancing coordination among all body parts.

## F DETAILS OF MULTI-MODAL BENCHMARK

To effectively evaluate our proposed task, we developed a multi-modal benchmark comprising 1,000 test clips, following the methodology outlined in Petrovich et al. (2024). Each test clip is automatically generated and includes two text descriptions and two audio clips.

For the text descriptions, we manually curated a set of 40 texts focusing on lower-body movements commonly associated with speech delivery or conversation. These descriptions provide the necessary context for evaluating the corresponding movements within the benchmark. Regarding the audio clips, we selected recordings from the BEATX dataset, specifically choosing eight speakers with speaker IDs below 10. These audio files were segmented into clips using a Voice Activity Detector (VAD), resulting in 694 audio clips with an average duration of 9.14 seconds.

The 1,000 test clips were generated through an automated process that utilizes the curated text descriptions and audio clips. For each test clip, two text descriptions are randomly selected and assigned random durations. Subsequently, two neighboring audio clips are randomly chosen. The start times for both the text and audio intervals are determined randomly, allowing the sequence to commence with either text or audio. This process results in the creation of four intervals that correspond to the selected text descriptions and audio clips.

Optional text descriptions are listed in the next page.

walk in a circle clockwise
walk in a circle counterclockwise
walk in a quarter circle to the left
walk in a quarter circle to the right
turn 180 degrees to the left on the left foot
turn 180 degrees to the left on the right foot
turn left
turn right
walk forwards
walk backwards
slowly walk forwards
slowly walk backwards
quickly walk forwards
quickly walk backwards
run
jogs forwards
jogs backwards
slowly walk in a circle
perform a squat
sit down
turn around then sit down in a chair
sit down then get back up and walk back
sit down for a moment
step back and sit down
sit down indian style
take a step to their right and sit down
sit criss cross
sit down on the ground and cross their legs
squat down
sit on a high object
sit on a barstool and rest their legs on the stool
take a large step and sits on a stool
get down on their knees
sit on the ground with his legs extended in front of him
walk up to a backwards chair and sit down on it with legs outstretched
sit down and adjust themselves
sit down and swap their legs crossing back and forth
sit and lie down on a lounge chair
sit down and lean on the chair
sits very still in the chair

