# OpenReview forum: "Multi-modal Controlled Coherent Motion Synthesis"
_ICLR.cc/2025/Conference — Submitted to ICLR 2025_

### Official Review · Reviewer_FZJK · 2024-11-02

**Soundness:** 4
**Presentation:** 3
**Contribution:** 4
**Rating:** 8
**Confidence:** 4

**Summary:**

The paper presents a diffusion-based framework aimed at generating realistic and coherent 3D avatar motions driven by concurrent multimodal inputs, such as text descriptions and speech audio. The key innovation of MOCO lies in its decoupled motion generation process, which independently produces motions for each modality from input noise and assembles body parts according to predefined spatial rules. This iterative approach enables each modality to refine its contribution within the context of the overall motion, resulting in increasingly natural and fluid movements that achieve coherence and synchronization across modalities.

The authors tackle the challenge of aligning motions with multiple control signals by proposing a method that simultaneously processes speech audio, text descriptions, and trajectory data, eliminating the need for additional datasets. The MOCO framework is trained on various datasets to ensure it can independently generate motions conditioned on either text or speech inputs. At each denoising step, the model generates distinct motions for the upper and lower body, conditioned on speech and text, respectively, and combines them to produce a cohesive motion that is then diffused for the next iteration. Additionally, the paper introduces a multi-modal benchmark for evaluating the proposed method.

**Strengths:**

The paper introduces a diffusion-based framework for multi-modal controlled motion synthesis, characterized by its innovative approach to decoupling motion generation.
- This decoupling facilitates the independent processing of speech audio, text descriptions, and trajectory data, effectively addressing a significant gap in the field where aligned multi-modal data is limited.
- The authors develop a purpose-built multimodal benchmark that enhances the validity of their claims.
- The paper is well-structured and clearly articulated. The introduction presents a compelling motivation.

**Weaknesses:**

**Performance in Single-Modality Scenarios:** The paper does not provide a direct comparison of MOCO's performance when operating in single-modality scenarios (text-only or audio-only) against existing text-to-motion (T2M) and audio-to-gesture (A2G) methods. While MOCO is designed for multi-modal input, discussing its performance in these more constrained scenarios is crucial.

**Conflict Resolution in Multi-Modal Control:** The paper describes a decoupled approach where audio primarily influences upper-body motion and text influences lower-body motion. However, it does not explicitly address how conflicts between modalities are resolved, particularly for the upper body. For instance, if text suggests lifting the left hand while audio cues indicate hand gestures, it is unclear how MOCO would reconcile these instructions. The paper would benefit from a more detailed discussion on conflict resolution strategies and potential limitations this might introduce. Experimental analysis demonstrating how such conflicts are handled and the impact on motion coherence would strengthen the paper's contributions.

**Specificity in Upper Body Motion Control:** While the decoupled approach is innovative, the paper lacks specificity on how upper body motion is controlled when both audio and text are provided. It is not detailed whether the model prioritizes one modality over the other in case of conflict, or if there is a blending strategy that integrates both inputs effectively. Clarifying the control mechanism and its implications on motion realism and coherence is essential for a comprehensive understanding of MOCO's capabilities and limitations.

**Limited visualization results:** Only one demonstration case is provided in the supplementary material. It is recommended to present more generated results as animations to visually assess the generalization ability of the proposed method.

**Questions:**

- Could the authors provide a comparison of MOCO's performance in text-only and audio-only scenarios with existing T2M and A2G methods?
- How does MOCO address situations in which text and audio provide contradictory motion cues for the same body part? Could the authors provide some examples to illustrate this?
- If users wish to control upper body motions using text, how flexible is MOCO's framework in accommodating such requirements? Can the authors elaborate on the potential modifications needed and any trade-offs involved? How does the framework integrate or prioritize different inputs when there is a mismatch?

---

> ### Author Response · Authors · 2024-11-26
>
> 1. **Performance on Single-Modality Conditions**
>    Thank you for your valuable advice. As suggested, we have conducted experiments to evaluate the performance of MOCO in single-modality conditions, as presented in Tables R1 and R2. Specifically, we trained MOCO from scratch on HumanML3D for text-to-motion and on BEATX for speech-to-gesture, respectively, ensuring a fair comparison.
>
>    | Methods                         | Top 1$\uparrow$| Top 2$\uparrow$| Top 3$\uparrow$| FID$\downarrow$      | MM Dist$\downarrow$ | Diversity$\uparrow$  | MM$\uparrow$         |
>    |---------------------------------|----------------------|----------------------|----------------------|----------------------|---------------------|----------------------|----------------------|
>    | Ground Truth                    | $0.511^{\pm .003}$   | $0.703^{\pm .003}$   | $0.797^{\pm .002}$   | $0.002^{\pm .000}$   | $2.974^{\pm .008}$  | $9.503^{\pm .065}$   | -                    |
>    | T2M-GPT                          | $0.491^{\pm .003}$   | $0.680^{\pm .003}$   | $0.775^{\pm .002}$   | $0.116^{\pm .004}$   | $3.118^{\pm .011}$  | $9.761^{\pm .081}$   | $1.856^{\pm .011}$   |
>    | MDM                              | -                    | -                    | $0.611^{\pm .007}$   | $0.544^{\pm .044}$   | $5.566^{\pm .027}$  | $9.559^{\pm .086}$   | $2.799^{\pm .072}$   |
>    | **MOCO (Ours)**               | $0.434^{\pm .010}$   | $0.618^{\pm .008}$   | $0.720^{\pm .008}$   | $0.530^{\pm .044}$   | $3.563^{\pm .049}$  | $9.856^{\pm .166}$   | $2.663^{\pm .068}$   |
>    | FineMoGen                        | $0.504^{\pm .002}$   | $0.690^{\pm .002}$   | $0.784^{\pm .002}$   | $0.151^{\pm .008}$   | $2.998^{\pm .008}$  | $9.263^{\pm .094}$   | $2.696^{\pm .079}$   |
>    | MoMask                           | $0.521^{\pm .002}$   | $0.713^{\pm .002}$   | $0.807^{\pm .002}$   | $0.045^{\pm .002}$   | $2.958^{\pm .008}$  | -                    | $1.241^{\pm .040}$   |
>    | LMM-Tiny                         | $0.496^{\pm .002}$   | $0.685^{\pm .002}$   | $0.785^{\pm .002}$   | $0.415^{\pm .002}$   | $3.087^{\pm .012}$  | $9.176^{\pm .074}$   | $1.465^{\pm .048}$   |
>    | LMM-Large                        | $0.525^{\pm .002}$   | $0.719^{\pm .002}$   | $0.811^{\pm .002}$   | $0.040^{\pm .002}$   | $2.943^{\pm .012}$  | $9.814^{\pm .076}$   | $2.683^{\pm .054}$   |
>
>    **Table R1**: Quantitative results of text-to-motion generation on the HumanML3D test set.
>
>    | Methods                         | FGD↓  | BC    | Diversity↑ | MSE↓  | LVD↓  |
>    |---------------------------------|-------|-------|------------|-------|-------|
>    | FaceFormer                      | -     | -     | -          | 7.787 | 7.593 |
>    | CodeTalker                      | -     | -     | -          | 8.026 | 7.766 |
>    | S2G                              | 28.15 | 4.683 | 5.971      | -     | -     |
>    | Trimodal                         | 12.41 | 5.933 | 7.724      | -     | -     |
>    | HA2G                             | 12.32 | 6.779 | 8.626      | -     | -     |
>    | DisCo                            | 9.417 | 6.439 | 9.912      | -     | -     |
>    | CaMN                             | 6.644 | 6.769 | 10.86      | -     | -     |
>    | DiffStyleGesture                 | 8.811 | 7.241 | 11.49      | -     | -     |
>    | TalkShow                         | 6.209 | 6.947 | 13.47      | 7.791 | 7.771 |
>    | EMAGE                            | 5.512 | 7.724 | 13.06      | 7.680 | 7.556 |
>    | ProbTalk                         | 6.170 | 8.099 | 10.43      | 8.990 | 8.385 |
>    | **MOCO (Ours)**                 | 5.543 | 7.089 | 14.05      | 7.285 | 7.573 |
>
>    **Table R2**: Quantitative results of speech-to-gesture generation on the BEATX test set.
>
>    In the HumanML3D text-to-motion benchmark (Table R1), our model achieves performance comparable to the widely-used MDM. This outcome is expected since our text-to-motion denoiser, $G_{\text{T2M}}$, is based on MDM. In the BEATX speech-to-gesture benchmark (Table R2), MOCO attains competitive performance compared to state-of-the-art methods.

---

> ### Author Response · Authors · 2024-11-26
>
> 2. **Addressing Contradictory Motion Cues**
>    Thank you for your valuable advice. Our method is designed based on the observation that, during speech, the audio signal typically provides sufficient information to drive the upper body motion. Users, therefore, only need to provide text descriptions to control the lower body movements. However, as the reviewer pointed out, there are scenarios where users may wish to control specific upper body motions, such as waving hands while talking. To clarify how our method addresses this scenario, we provide the following example:
>
>    Given the multi-modal conditions, our method first creates a timeline based on the provided conditions and then generates the corresponding motion according to this timeline. For example:
>
>    ```
>    a man walks then waves hands # 0.0 # 8.0 # legs # spine
>    speak:1_wayne_0_103_103,$9248$99808 # 2.578 # 8.238   # left arm # right arm # head
>    speak:1_wayne_0_103_103,$107552$196064 # 9.722 # 15.254   # left arm # right arm # head
>    ```
>
>       **Explanation**:
>
>    - **Text Description**: "a man walks then waves hands"
>
>      - **Start and End Time in Overall Motion**: `# 0.0 # 8.0`
>      - **Controlled Body Parts**: `# legs # spine`
>
>    - **Audio Input**: `speak:1_wayne_0_103_103`
>      - **Start and End Frames in Audio File**: `$9248$99808`
>      - **Start and End Time in Overall Motion**: `# 2.578 # 8.238`
>      - **Controlled Body Parts**: `# left arm # right arm # head`
>
>       **Explanation**:
>
> 	In this example, a conflict arises between the upper body motion cues from "waves hands" (text description) and the speech audio during the overlapping period of 2.578 seconds to 8.0 seconds. To resolve such conflicts, our method relies on a predefined timeline, which gives precedence to the speech audio in governing the upper body motion during this period. Consequently, the "waves hands" motion is suppressed between 2.578 and 8.0 seconds.
>
>    While this illustrates a limitation of our approach, the flexibility of MOCO provides a potential solution to address such conflicts by allowing finer-grained user control.
>
>    Additionally, we have added details of multi-modal control rules in **Appendix G**.
>
> 3. **Flexibility of MOCO**
>
>    Thank you for your valuable advice. Using the example provided earlier, we demonstrate the flexibility of MOCO as follows:
>
>    ```
>    a man walks then waves hands # 0.0 # 8.0 # legs # spine
>    speak:1_wayne_0_103_103,$9248$99808 # 2.578 # 8.238   # left arm # right arm # head
>    speak:1_wayne_0_103_103,$107552$196064 # 9.722 # 15.254   # left arm # right arm # head
>    ```
>    In this scenario, the audio governs the upper body motion during the contradictory period. However, **if users wish to control upper body motions using text during speech**, we can simply adjust the timeline as follows to achieve this:
>
>    ```
>    a man walks # 0.0 # 8.0 # legs # spine
>    wave hands # 7.0 # 11.0 # left arm # right arm
>    speak:1_wayne_0_103_103,$9248$99808 # 2.578 # 8.238   # left arm # right arm # head
>    speak:1_wayne_0_103_103,$107552$196064 # 9.722 # 15.254   # left arm # right arm # head
>    ```
>
> 	By modifying the timeline, the text command "wave hands" now controls the arms from 7.0 to 11.0 seconds. Since the speech audio spans from 2.578 to 15.254 seconds, there is an overlap between 7.0 and 11.0 seconds where both audio and text attempt to control the arms. According to our conflict resolution rules, the condition with fewer control parts takes precedence. In this case, the text command "wave hands" governs the arms during the overlapping period.
>
> 	As a result, users can control upper body motions using text while speech audio is active. The corresponding visualization is provided in the supplementary materials as **flexibility.mp4**.
>
> 	This demonstrates the flexibility of our method. Even though our default configuration prioritizes audio for the upper body and text for the lower body, users can easily reconfigure these priorities through timeline adjustments to achieve their desired results.
>
> 4. **Visualization Results**
>
>    Thank you for your valuable advice. As suggested, we have added more video samples to the supplementary material in the **Qualitative Results** folder to visually demonstrate MOCO's performance.

---

> > ### Comment · Reviewer_FZJK · 2024-11-28
> >
> > Thank you for the detailed feedback. I believe the exploration of this work is valuable for the community and has great potential for application. Therefore, I have raised my score from 6 to 8. I hope the authors release their code as promised.

---

> > > ### Author Response · Authors · 2024-11-28
> > >
> > > Thank you once again for your valuable insights! We are committed to releasing the code as soon as possible.

---

### Official Review · Reviewer_eouV · 2024-11-03

**Soundness:** 3
**Presentation:** 3
**Contribution:** 2
**Rating:** 6
**Confidence:** 4

**Summary:**

This work addresses a very specific problem in 3D human motion— speech generation, which involves generating both body movement and detailed facial and hand gestures conditioned on text , speech audio and trajectory. Specifically, the proposed method uses four models to achieve (1) text to lower body motion, (2) speech to upper body motion, (3) speech audio to facial and hand details, and (4) trajectory to global velocity, to synthesize actions for walking and talking simultaneously. Overall, the method is clear, and achieves state-of-the-art results in the benchmark made by mixed HumanML3D and BEATX.

However, I feel that the work’s novelty might be somewhat limited. This work focuses on a specific application. Although it could be useful for speech generation, it doesn’t seem to demonstrate other areas of extensibility. Methodologically, it seems more like a combination and application of existing models, without considering deeper connection between speech content and body movement.

--------
The authors provided experiments that use LLM as a high level guider/coodinator. These experiments seem interesting and inspiring, and they show a potential to broader and easier use of this method. Therefore I raise the score from 5 to 6.

**Strengths:**

+ The problem of speech motion generation is interesting and has application value (although it may seem somewhat overly specific for a machine learning conference).

+ The writing is clear. Each part of the proposed method is reasonable.

+ The experiments are thorough. The benchmark is well-considered, addressing both movement (HumanML3D) and detailed gestures (BEATX). The ablation study is also comprehensive.

**Weaknesses:**

- Although the method is reasonable, it still appears to be a combination of different body parts. These combinations sometimes don’t look entirely natural. For instance, around the 23 second in the demo, when the human starts moving, the upper and lower body seem somewhat unnaturally separate. Normally, human upper and lower movements have some interrelated dynamics (based on physical balance or habit), which I didn't see the authors address in the proposed solution.

- The proposed method seems to rely on precise, pre-arranged text and speech audio clips rather than handling this autonomously. Specifically, given a speech content, the method does not help users plan when to say what or when to perform certain actions, which could limit its application scenarios. Perhaps the authors could consider using LLMs to assist with some high-level planning or text-conditioned generation? I believe doing so would not add too much workload but could make the method much fancier.

- There appears to be a lack of semantic-level coherence between the speech content and the motion. In the demo, gestures seem to change with the rhythm of the speech, but the motion itself seems to lack meaning or understanding. Again, have the authors considered using language models to generate movement details that are more closely aligned with the content of the speech?

- This work involves different modality conditions, one of which is audio to motion. However, some important references on "audio to motion," such as dance generation (e.g., Bailando, CVPR 2022, and EDGE, CVPR 2023), seem to be missing.

**Questions:**

See weakness.

---

> ### Author Response · Authors · 2024-11-26
>
> 1. **Improvements on Whole-body Naturalness**
>
> 	Thank you for your valuable advice. Our methods have made efforts and achieved improvements in whole-body naturalness. Although the method has room for optimization, we believe this exploration is valuable and can inspire further research into multi-modal conditional generation, particularly in challenging domains where paired datasets are difficult to obtain.
>
> 	Specially, we apply the decouple-then-combine strategy at each diffusion step, ensuring that each denoiser generates the corresponding body parts with an awareness of the entire body motion. The corresponding theoretical analysis is presented in Appendix A. Additionally, we train the text-to-motion denoiser using additional speech gesture data labeled with pseudo-text such as “a man is speaking.” This training strategy aims to enhance the text-to-motion denoiser’s ability to generate body movements that are coordinated with speech gestures.
>
> 	To illustrate this more clearly, we have added a visualization comparison between MOCO and a baseline, named **Combine-Only_Last_Step**, to the supplementary material in the **Qualitative Results** folder. The baseline **Combine-Only_Last_Step** employs the combine strategy only in the final step and does not utilize pseudo-labeled speech gesture data to train the text-to-motion denoiser. The text descriptions for this comparison, shown in **video_1.mp4**, are “jogs forwards” and “walks in a circle counterclockwise.” It is noted that MOCO generates lower body motions that are more coordinated with speech gestures. This is because fast running typically requires arm swinging, whereas the arms in speech gestures do not swing side to side. To accommodate this, the lower body motion generated by MOCO has a smaller amplitude and maintains an upright spine to ensure stability. In contrast, the baseline generates lower body motions with larger amplitudes and a bent spine, but the arm movements are stiff and inconsistent with the substantial lower body movements, resulting in unrealistic overall motion. This demonstrates that our method achieves better coordination of upper and lower body dynamics compared to the baseline.
>
> 	However, as pointed out by the reviewer, the motion generated by MOCO has certain artifacts. We will continue to improve whole-body coordination and dynamics in future work. One approach is to incorporate larger training datasets, such as music-to-dance datasets, to improve whole-body dynamics. Another approach is to train the speech-to-gesture denoiser with a text-to-motion dataset to ensure that the generated speech gestures are better coordinated with lower body movements.
>
> 	Additionally, we have included more visual comparisons between MOCO and **Combine-Only_Last_Step** to highlight the improvements MOCO achieves in whole-body naturalness. For details, please refer to the **Qualitative Results** folder in the supplementary materials.
>
> 2. **LLM-Based Motion Planning (Part I)**
>
> 	Thank you for your valuable advice. Our motion generation framework, **MOCO**, is designed to produce motion sequences based on predefined timelines. Below, we present the timeline formats utilized for text and audio driven motions:
>
> 	- **Text:** `description # start time # end time # involved body parts`
> 	- **Audio:** `audio name $audio start Hz$audio end Hz # start time # end time # involved body parts`
>
>
> 	Example:
> 	```
> 	quickly walk backwards # 0.0 # 4.46 # legs # spine
> 	walk in a quarter circle to the left # 4.46 # 8.67 # legs # spine
> 	speak:4_lawrence_0_95_95,$29728$147424 # 0.0 # 7.35 # head # left arm # right arm
> 	speak:4_lawrence_0_95_95,$151584$255456 # 7.35 # 13.84 # head # left arm # right arm
> 	```
>
> 	To enable the **LLM** to generate timelines automatically based on the given audio information, we follow these structured steps:
>
> 	1. **Introduce the Timeline Format:**
>        Begin by explaining the structure and syntax of the timeline, detailing how start and end times, body parts, and additional information are represented.
>
> 	2. **Provide Examples:**
> 	Share a few illustrative examples to demonstrate the expected output format, ensuring clarity on how timelines should be generated.
>
> 	3. **Present Optional Text Descriptions:**
>        Offer a set of potential text descriptions that can be used to guide the motion generation process alongside the audio data.
>
> 	4. **Present Audio Information:**
>        Provide the audio details, including the audio name and the corresponding start and end frequencies. For instance:
>        - `speak:5_stewart_0_87_87,$630304$713696`
>        - `speak:5_stewart_0_87_87,$718368$807392`

---

> ### Author Response · Authors · 2024-11-26
>
> 2. **LLM-Based Motion Planning (Part II)**
>
> 	Based on the inputs provided, the LLM generates timelines such as:
> 	```
> 	walk forwards # 0.00 # 5.21 # legs # spine
> 	speak:5_stewart_0_87_87,$630304$713696 # 0.00 # 5.21 # head # left arm # right arm
> 	sit down for a moment and then stand up # 5.21 # 10.77 # legs # spine
> 	speak:5_stewart_0_87_87,$718368$807392 # 5.21 # 10.77 # head # left arm # right arm
> 	```
>
> 	For detailed conversation logs and additional context, please refer to **LLM_motion_planning** in the supplementary materials.
>
> 3. **LLM Boost Semantic-Alignment (Part I)**
>
> 	Thank you for your valuable insights. Utilizing large language models (LLMs) to enhance semantic motion, particularly semantic gestures, is promising and aligns with our future objectives. For example, when a speaker discusses a specific topic, LLMs can interpret the context and generate controls to perform corresponding motions. MOCO can facilitate the realization of this concept. However, our experiments indicate that achieving such precise control requires further effort, including designing effective prompts, expanding text-to-hand datasets, and bridging audio-to-semantic processing using LLMs. Addressing these challenges will be essential for advancing our capabilities in semantic motion generation.
>
> 	As suggested, we have tested the performance of several well-known LLMs on this task, including GPT-4o-mini, GPT-4o, GPT-o1-mini, GPT-o1-preview, Claude3.5-sonnet, and Gemini-1.5 pro. Notably, since these advanced models typically lack the capability to handle audio modalities, we simplified the scenario by manually providing the speech content. For example:
> 	```
> 	speak:5_stewart_0_87_87,$630304$713696, the speech content is about a man discussing the weather
> 	speak:5_stewart_0_87_87,$718368$807392, the speech content is about a man expressing excitement
> 	```
> 	Please note that the actual content of the speech audio does not correspond to the content we have manually provided. It is intended solely for testing purposes. Inspired by the valuable insights from **Reviewer FZJK**, we encourage LLMs to use arm gestures to express emotions or other aspects aligned with the speech content. The timelines generated by the LLMs are listed below:
> 	```
> 	# GPT-4o-mini (incorrect audio duration)
> 	a man stands and looks around # 0.0 # 5.0 # head # spine
> 	speak:5_stewart_0_87_87,$630304$713696 # 1.0 # 6.0 # head # left arm # right arm
> 	gesture towards the sky # 5.0 # 10.0 # left arm # right arm # head
> 	speak:5_stewart_0_87_87,$718368$807392 # 8.0 # 13.0 # head # left arm # right arm
> 	jump with excitement # 10.0 # 15.0 # legs # spine
>
> 	# GPT-4o (missing condition at 5.5s - 6.0s)
> 	speak:5_stewart_0_87_87,$630304$713696 # 0.0 # 5.222 # left arm # right arm # head
> 	gesture thoughtfully # 0.5 # 5.5 # left arm # right arm # head
> 	speak:5_stewart_0_87_87,$718368$807392 # 6.0 # 11.563 # left arm # right arm # head
> 	jump with joy # 6.5 # 12.0 # legs # spine # left arm # right arm
>
> 	# GPT-o1-mini (missing condition at 5.212s - 5.3s)
> 	speak:5_stewart_0_87_87,$630304$713696 # 0.0 # 5.212 # left arm # right arm # head
> 	nod head and gesture with arms # 0.1 # 5.1 # left arm # right arm # head
> 	speak:5_stewart_0_87_87,$718368$807392 # 5.3 # 10.864 # left arm # right arm # head
> 	jump up and down with arms raised # 5.4 # 10.4 # left arm # right arm # legs # spine
>
> 	# GPT-o1-preview (missing condition at 0.0s - 0.5s)
> 	Gesture about the weather # 1.0 # 6.0 # left arm
> 	Walk forward slowly # 1.1 # 6.1 # legs # spine
> 	speak:5_stewart_0_87_87,$630304$713696 # 0.5 # 5.712 # left arm # right arm # head
> 	Raise hands in excitement # 6.7 #11.7 # left arm # right arm
> 	Walk energetically # 6.4 #11.4 # legs # spine
> 	speak:5_stewart_0_87_87,$718368$807392 #6.3 #11.864 # left arm # right arm # head
>
> 	# Claude3.5-sonnet
> 	standing casually # 0.0 # 5.0 # legs # spine
> 	look at sky # 0.2 # 5.2 # head
> 	gesture at clouds # 2.0 # 7.0 # left arm # right arm
> 	speak:5_stewart_0_87_87,$630304$713696 # 0.0 # 5.212 # left arm # right arm # head
> 	jump excitedly # 5.3 # 10.3 # legs # spine
> 	raise arms enthusiastically # 7.2 # 12.2 # left arm # right arm
> 	nod vigorously # 7.4 # 12.4 # head
> 	speak:5_stewart_0_87_87,$718368$807392 # 5.564 # 11.128 # left arm # right arm # head
>
> 	# Gemini-1.5 pro (missing condition at 10.5s - 11.0s)
> 	walking casually # 0.0 # 10.0 # legs # spine
> 	speak:5_stewart_0_87_87,$630304$713696 # 4.451 # 8.903 # left arm # right arm # head
> 	glance at the sky # 6.0 # 7.5 # head
> 	shrug shoulders # 9.0 # 10.5 # left arm # right arm # head
> 	speak:5_stewart_0_87_87,$718368$807392 # 11.0 # 15.453 # left arm # right arm # head
> 	clap hands # 13.0 # 14.0 # left arm # right arm
> 	jump excitedly # 14.5 # 16.0 # legs # spine
> 	point forward # 15.5 # 16.5 # right arm
> 	```

---

> ### Author Response · Authors · 2024-11-28
>
> 3. **LLM Boost Semantic-Alignment (Part II)**
>
> 	We observed that most LLMs generate incorrect timelines, such as gaps without conditions or inaccurate calculations of audio durations. Subsequently, we manually corrected the incorrect time configurations and minor conflicts, then visualized all results. Detailed conversation logs and visualization results are available in the supplementary materials (**LLM-boost-semantic** folder).
>
> 	From the visualization results, it can be observed that MOCO accurately generates semantic-aligned motions according to LLM instructions, such as "Raise hands in excitement," "gesture at clouds," "jump excitedly," and "clap hands," seamlessly integrating with conversational gestures. However, for more fine-grained motions, such as "clap hands," the absence of a comprehensive text-to-hand-motion dataset limits the motion to arm movements only. Additionally, Gemini-1.5 pro generates too many short-term motions, for example,``point forward # 15.5 # 16.5,`` which poses challenges in achieving such precise timing for motions.
>
> 	Despite these minor limitations, we found that using LLMs to enhance the semantic alignment between motion and speech content is feasible and promising. We thank you once again for the valuable insights. Due to time constraints during the rebuttal period, we will include detailed experiment information in the revised version.
>
> 4. **Important References**
>
> 	Thank you for your valuable advice. These works are indeed significant in this field. As suggested, we have added these important references in the related work section. Furthermore, we will incorporate additional relevant studies to enhance the comprehensiveness of our literature review.

---

> > ### Comment · Reviewer_eouV · 2024-12-01
> >
> > Thank you for your great effort in rebuttal. After reading the revised content I feel a large part of my concern has been addressed. Therefore I raised the score.
> >
> > LLM provides a good way for generation and editing. I think the authors may try exploring more on this to extend it to broader and easier usage.

---

> ### Author Response · Authors · 2024-12-03
>
> Thank you for your insightful feedback and for the effort helping us improve the paper. We are pleased to hear that a large part of your concern has been addressed.
>
> We also appreciate your suggestion regarding the potential of LLMs for generation and editing. We recognize the value of exploring LLMs to further enhance the capabilities of our approach, particularly in terms of enabling broader and more user-friendly applications. This is an exciting direction for future work, and we plan to explore bridging LLMs with multi-modal conditional control for motion generation—incorporating LLMs to understand multi-modal conditions and effectively perform motion planning.
>
> Once again, thank you for your valuable input and constructive suggestions.

---

### Official Review · Reviewer_itKM · 2024-11-04

**Soundness:** 3
**Presentation:** 3
**Contribution:** 3
**Rating:** 5
**Confidence:** 3

**Summary:**

The paper introduces MOCO, a novel diffusion-based framework for 3D avatar motion synthesis that effectively integrates multi-modal inputs such as text descriptions, speech audio, and trajectory data. By independently modeling each modality using transformer-based denoisers and employing a decoupled denoising process, MOCO generates fluid and coherent motion sequences. The framework splits motion generation into upper-body movements driven by speech and lower-body movements influenced by text, refining each component iteratively. It also incorporates trajectory data to guide global body transitions and manages asynchronous conditions with a timeline-based strategy. Experiments on datasets like HumanML3D, BEATX, and a custom multi-modal benchmark demonstrate that MOCO outperforms baseline models, achieving superior synchronization and coherent motion. The paper suggests potential improvements to address minor limitations such as foot sliding during transitions.

**Strengths:**

Originality: The paper presents a novel framework that effectively combines multiple modalities for motion synthesis, addressing a significant gap in existing research.
Quality: The decoupled denoising approach is well-designed, allowing for independent modeling of upper-body and lower-body movements, which enhances the coherence and synchronization of generated motions.
Clarity: The methodology is clearly explained, with detailed descriptions of the transformer-based denoisers and how they interact within the framework.
Significance: The ability to synthesize lifelike motion sequences from multi-modal inputs has important implications for virtual avatars, gaming, and animation industries.

**Weaknesses:**

Computational Complexity: The paper lacks a discussion on the computational efficiency and scalability of the proposed framework, which is crucial for real-time applications.
Limited Ablation Studies: While an ablation study is mentioned, more extensive experiments isolating the contributions of each component could strengthen the evaluation.
Dataset Diversity: The datasets used may not cover all possible real-world scenarios, potentially limiting the generalizability of the results.
Minor Artifacts: The issue of foot sliding during transitions is acknowledged but not thoroughly addressed, leaving room for improvement in motion realism.

**Questions:**

How does MOCO perform in terms of computational efficiency, especially in real-time applications?
Can the authors provide more details on how the decoupled denoising approach affects the overall computational complexity?
Have the authors considered the integration of additional modalities, such as facial expressions or environmental context?
What strategies could be employed to mitigate the foot sliding issue during motion transitions?
How generalizable is MOCO to scenarios beyond the datasets used, particularly in more complex or varied environments?

**Details Of Ethics Concerns:**

No ethics review is needed.

---

> ### Author Response · Authors · 2024-11-26
>
> 1. **Computational Complexity**
>    Thank you for your valuable advice. As suggested, we present various metrics, including the number of parameters, model size, FLOPs, and inference time of MOCO on a single NVIDIA 4090 GPU, as shown in Table R1. As indicated in the table, each denoiser in our framework achieves an inference speed of less than 10 ms per frame.
>
>    |                            | Parameters (M) | Model Size (MB) | FLOPs (G) | Inference Time (ms/frame) |
>    |----------------------------|----------------|-----------------|-----------|---------------------------|
>    | $G_{\text{T2M}}$            | 27.01          | 103.02          | 5.19      | 2.26                      |
>    | $G_{\text{S2G}}$            | 36.86          | 140.62          | 6.72      | 4.30                      |
>    | $G_{\text{T2V}}$            | 0.34           | 1.31            | 0.06      | 6.20                      |
>    | $G_{\text{S2D}}$            | 36.94          | 140.94          | 6.74      | 4.37                      |
>
>    **Table R1**: Complexity of each denoiser in MOCO.
>
>    Considering that actual testing involves numerous additional operations—such as reassembling vectors after decoupled inference and routing the corresponding parts to their respective denoisers—we demonstrate MOCO's inference speed with a practical example. To generate the motion sequences for a 35-second demo video, which consists of nine clips under different conditions and has a total duration of 54 seconds, our method completed the body motion generation task in only 3.72 seconds. This rapid generation time highlights the potential of our approach for real-time applications.
>
> 2. **Ablation Studies**
>    Thank you for your valuable advice. In the original manuscript, we conducted ablation studies on key design aspects, including multi-modal condition methods (e.g., weighted average and pseudo-text, as presented in Manuscript Table 1), the sharing of weights between the text-to-motion denoiser and the speech-to-gesture denoiser, the body parts controlled by each modality, and the transition method (as shown in Manuscript Table 2). As suggested, we have performed additional experiments such as ablation study on trajectory control, comparison of MOCO in synchronous and asynchronous conditions, single-modality performance, and so on. Please refer to **Appendix E** for more details.
>
> 3. **Generalization.**
>
> 	Thank you for your valuable advice. As suggested, we have conducted experiments beyond controlled benchmarks to demonstrate how MOCO performs in more complex and varied environments. Specifically, in these challenging scenarios, we utilized six different LLMs to generate text descriptions and time configurations, constructing timelines based on the given audio content. We then employed MOCO to generate motions according to these timelines. The visualization results are presented in the **LLM-boost-semantic** folder within the supplementary materials.
>
> 	Additionally, we tested MOCO's robustness against varied environments by using in-the-wild audio. We sampled two audio clips from SHOW, a speech-to-gesture dataset collected from real-world talk show videos, and used MOCO to generate motions based on these audio clips. The visualization results for these tests are available in the **generalization_to_show** folder within the supplementary materials.
>
> 	We observed that MOCO can generally generate realistic and semantically aligned motions based on the conditions automatically generated by LLMs. Furthermore, MOCO demonstrates robustness against out-of-domain audio inputs. Overall, MOCO exhibits superior performance in more complex and varied environments.
>
> 4. **Solving Minor Artifacts**
>    Thank you for your valuable advice. We identified minor artifacts, such as foot sliding, in our method but have not thoroughly addressed them. These artifacts mainly occur when conditions from two modalities are provided simultaneously, and the text condition requires a significant change in the overall body motion state, for example, "stand up." We believe that the primary cause of this issue is the limited capability of the text-to-motion denoiser, which struggles to effectively handle situations where the upper body is controlled by speech gestures while the lower body undergoes such transitions. One potential solution is to use a larger dataset to pre-train the text-to-motion denoiser, such as Motion-X.

---

> ### Author Response · Authors · 2024-11-28
>
> 5. **Additional Modalities**
>    Thank you for your valuable advice. In our manuscript, we introduced facial expressions through the speech-to-details denoiser, which manages facial expressions and hand poses. Incorporating environmental context is much more challenging. Currently, we can predict trajectories in the environment using an off-the-shelf algorithm, such as A-star, and then generate actions within the scene based on trajectory control in MOCO. However, the interaction between these actions and the environment may be limited and imprecise. We will incorporate environmental context into the conditions in future work.

---

### Official Review · Reviewer_rsd5 · 2024-11-05

**Soundness:** 2
**Presentation:** 3
**Contribution:** 2
**Rating:** 5
**Confidence:** 3

**Summary:**

The paper introduces MOCO, a diffusion-based framework for generating 3D avatar motion driven by simultaneous multi-modal inputs, such as speech audio, text descriptions, and trajectory data. A novel decoupling mechanism that allows upper-body and lower-body motions to be generated separately and integrated cohesively is proposed, ensuring realistic behavior across modalities. Experiments on a purpose-built benchmark show that MOCO outperforms existing methods on generating synchronized and lifelike avatar motions.

**Strengths:**

1. MOCO introduces the first approach to handle simultaneous multi-modal inputs (speech, text, and trajectory) in motion synthesis, overcoming the limitations of sequential or averaged methods used in previous work.

2. The framework’s decoupling strategy, which independently processes each modality and then integrates them according to spatial rules, ensures high alignment and coherence across modalities, leading to realistic and synchronized avatar behavior.

**Weaknesses:**

1. Limited Real-World Testing: The experiments primarily focus on controlled benchmarks and do not demonstrate how MOCO performs in real-world or diverse environments, which limits understanding of its practical application.

2. Analysis of Failure Cases: The paper lacks an in-depth analysis of failure modes or specific situations where MOCO might struggle, such as handling overlapping or conflicting input conditions from different modalities.

3. The paper does not include user studies or subjective assessments to gauge the perceived naturalness and quality of the generated motions, which would add practical validation to the quantitative results.

4. Although MOCO excels at combining multiple modalities for coherent motion generation, its joint performance comes at the expense of not achieving the highest scores in any single domain.

5. The paper provides qualitative results but lacks detailed video examples or demonstrations of the generated outputs.

**Questions:**

1. While MOCO performs well across joint multi-modal metrics, it does not surpass single-modality baselines in their respective domains. Have you considered methods to optimize modality-specific contributions without compromising joint performance?

2. Have you considered conducting user studies or collecting subjective feedback to assess the perceived naturalness and coherence of MOCO’s generated motions? This would provide additional validation for the results.

3. Could you provide more examples or analysis of specific scenarios where MOCO struggles, such as when input modalities provide contradictory cues or when generating highly dynamic or complex movements?

4. Could you provide video samples to showcase MOCO's performance visually? This would be highly beneficial for evaluating the naturalness and coherence of the synthesized motions.

---

> ### Author Response · Authors · 2024-11-26
>
> 1. **Joint Perfromance and Single Modality Performace (Part I)**
>
> 	Thanks for this concern. We would like to clarify that MOCO's performance on two-modality conditions does not compromise its effectiveness on single-modality condition. This misunderstanding is due to the pseudo ground-truth (GT) we adopted for MOCO. Specifically, since MOCO integrates both text-driven motion and speech-driven gestures, its GT differs from those for using text-to-motion or speech-to-gesture alone. However, as the ground-truth for MOCO on two-modality conditions is missing, we have to adopt the GT for the single-modality condition as the `pseudo-GT’ for MOCO. This is the very reason why MOCO’s performance on two-modality conditions seems to be lower than that on single-modality condition, i.e. single-modality baselines.
>
>  	To strictly evaluate MOCO's performance with single-modality conditions, we conduct experiments for MOCO to perform text-to-motion and speech-to-gesture tasks, respectively, as presented in Tables R1 and R2. Specifically, we train MOCO from scratch on the HumanML3D dataset for text-to-motion and on the BEATX dataset for speech-to-gesture, respectively, ensuring a fair comparison.
>
>    | Methods                         | Top 1$\uparrow$| Top 2$\uparrow$| Top 3$\uparrow$| FID$\downarrow$      | MM Dist$\downarrow$ | Diversity$\uparrow$  | MM$\uparrow$         |
>    |---------------------------------|----------------------|----------------------|----------------------|----------------------|---------------------|----------------------|----------------------|
>    | Ground Truth                    | $0.511^{\pm .003}$   | $0.703^{\pm .003}$   | $0.797^{\pm .002}$   | $0.002^{\pm .000}$   | $2.974^{\pm .008}$  | $9.503^{\pm .065}$   | -                    |
>    | T2M-GPT                          | $0.491^{\pm .003}$   | $0.680^{\pm .003}$   | $0.775^{\pm .002}$   | $0.116^{\pm .004}$   | $3.118^{\pm .011}$  | $9.761^{\pm .081}$   | $1.856^{\pm .011}$   |
>    | MDM                              | -                    | -                    | $0.611^{\pm .007}$   | $0.544^{\pm .044}$   | $5.566^{\pm .027}$  | $9.559^{\pm .086}$   | $2.799^{\pm .072}$   |
>    | **MOCO (Ours)**               | $0.434^{\pm .010}$   | $0.618^{\pm .008}$   | $0.720^{\pm .008}$   | $0.530^{\pm .044}$   | $3.563^{\pm .049}$  | $9.856^{\pm .166}$   | $2.663^{\pm .068}$   |
>    | FineMoGen                        | $0.504^{\pm .002}$   | $0.690^{\pm .002}$   | $0.784^{\pm .002}$   | $0.151^{\pm .008}$   | $2.998^{\pm .008}$  | $9.263^{\pm .094}$   | $2.696^{\pm .079}$   |
>    | MoMask                           | $0.521^{\pm .002}$   | $0.713^{\pm .002}$   | $0.807^{\pm .002}$   | $0.045^{\pm .002}$   | $2.958^{\pm .008}$  | -                    | $1.241^{\pm .040}$   |
>    | LMM-Tiny                         | $0.496^{\pm .002}$   | $0.685^{\pm .002}$   | $0.785^{\pm .002}$   | $0.415^{\pm .002}$   | $3.087^{\pm .012}$  | $9.176^{\pm .074}$   | $1.465^{\pm .048}$   |
>    | LMM-Large                        | $0.525^{\pm .002}$   | $0.719^{\pm .002}$   | $0.811^{\pm .002}$   | $0.040^{\pm .002}$   | $2.943^{\pm .012}$  | $9.814^{\pm .076}$   | $2.683^{\pm .054}$   |
>
>    Table R1: Quantitative results of text-to-motion generation on the HumanML3D test set.
>
>    | Methods                         | FGD↓  | BC    | Diversity↑ | MSE↓  | LVD↓  |
>    |---------------------------------|-------|-------|------------|-------|-------|
>    | FaceFormer                      | -     | -     | -          | 7.787 | 7.593 |
>    | CodeTalker                      | -     | -     | -          | 8.026 | 7.766 |
>    | S2G                              | 28.15 | 4.683 | 5.971      | -     | -     |
>    | Trimodal                         | 12.41 | 5.933 | 7.724      | -     | -     |
>    | HA2G                             | 12.32 | 6.779 | 8.626      | -     | -     |
>    | DisCo                            | 9.417 | 6.439 | 9.912      | -     | -     |
>    | CaMN                             | 6.644 | 6.769 | 10.86      | -     | -     |
>    | DiffStyleGesture                 | 8.811 | 7.241 | 11.49      | -     | -     |
>    | TalkShow                         | 6.209 | 6.947 | 13.47      | 7.791 | 7.771 |
>    | EMAGE                            | 5.512 | 7.724 | 13.06      | 7.680 | 7.556 |
>    | ProbTalk                         | 6.170 | 8.099 | 10.43      | 8.990 | 8.385 |
>    | **MOCO (Ours)**                 | 5.543 | 7.089 | 14.05      | 7.285 | 7.573 |
>
>    Table R2: Quantitative results of speech-to-gesture generation on the BEATX test set.

---

> ### Author Response · Authors · 2024-11-27
>
> 1. **Joint Perfromance and Single Modality Performace (Part II)**
>
> 	In the HumanML3D text-to-motion benchmark (Table R1), our model achieves performance comparable to the widely-used MDM. This outcome is reasonable since our text-to-motion denoiser, G_"T2M" , is based on MDM. In the BEATX speech-to-gesture benchmark (Table R2), MOCO attains competitive performance compared to state-of-the-art methods. The above experiments justify that MOCO's performance on two-modality conditions does not compromise its effectiveness on single-modality condition.
>
> 2. **User Study.**
>
> 	Thank you for your valuable advice.  As suggested, we have conducted user study and the results are presented in the below table.
>
> 	| **Method**          | **Better Text Following (%)** | **Better Beat Synchronization (%)** |
> 	|---------------------|-------------------------------|--------------------------------------|
> 	| Neither             | 1.0                           | 13.0                                 |
> 	| Pseudo-Text         | 0.0                           | 12.5                                 |
> 	| MOCO (Ours)         | 99.0                          | 74.5                                 |
>
> 	| **Method**          | **Better Text Following (%)** | **Better Beat Synchronization (%)** |
> 	|---------------------|-------------------------------|--------------------------------------|
> 	| Neither             | 0.0                           | 13.5                                 |
> 	| Weighted Average    | 0.0                           | 3.5                                  |
> 	| MOCO (Ours)         | 100.0                         | 83.0                                 |
>
> 	| **Method**               | **Better Body Coherence (%)** | **Better Temporal Fluidity (%)** |
> 	|--------------------------|-------------------------------|-----------------------------------|
> 	| Neither                  | 12.0                          | 12.6                              |
> 	| Combine Only Last Time   | 17.0                          | 42.6                              |
> 	| MOCO (Ours)              | 71.0                          | 44.8                              |
>
> 	Specifically, we evaluate MOCO against *Pseudo-Text* and *Weighted Average*, introduced in Section 4.3, to assess overall performance in text following and audio beat synchronization. Additionally, we compare MOCO with *Combine Only Last Time*, which also employs the decoupling strategy but applies the combining strategy only at the final diffusion step. This comparison aims to evaluate whole body coherence and temporal fluidity.
>
> 	As shown in the table, MOCO achieves significant advantages over both *Pseudo-Text* and *Weighted Average*, demonstrating the effectiveness of our decouple-then-combine strategy in generating motion aligned with multi-modal conditions. Furthermore, when compared to *Combine Only Last Time*, our method was rated significantly higher in both body coherence and temporal fluidity. This indicates that MOCO does more than merely combine different body parts controlled by separate conditions; it ensures that each body part aligns with its corresponding condition while enhancing coordination among all body parts.

---

> ### Author Response · Authors · 2024-11-28
>
> 3. **Analysis of Failure Cases.**
>
>     Thank you for your valuable advice. As suggested, we have added more video examples to the supplementary material that compare the qualitative performance of MOCO against baseline methods. Based on these video examples, it can be observed that MOCO generates more accurate text-following motions compared to the baseline methods.
>
>     However, we also acknowledge certain limitations of MOCO in these results. As shown in **limitation_1.mp4**, the most obvious limitation is that when conditions from two modalities are provided simultaneously, and the text condition requires a significant change in the overall body motion state—such as "sit down"—the generated motion appears too fast and results in foot sliding. This issue arises from the limited capability of the text-to-motion denoiser, which struggles to effectively handle situations where the upper body is controlled by speech gestures while the lower body undergoes such transitions. We plan to address this in our future work.
>
>     Another limitation, as pointed out by the reviewer, happens when when input modalities provide contradictory cues. Our method is designed based on the observation that, during speech, the audio signal typically provides sufficient information to drive the upper body motion. Users, therefore, only need to provide text descriptions to control the lower body movements. However, there are scenarios where users may wish to control specific upper body motions, such as "waving hands". In this scenario, audio will govern the upper body, and the text control would be omitted.
>
>     However, due to the flexibility of our method, MOCO can handle such contradictory cues simply adjusting the input timeline. For example:
>     ```
>    a man walks then waves hands # 0.0 # 8.0 # legs # spine
>    speak:1_wayne_0_103_103,$9248$99808 # 2.578 # 8.238   # left arm # right arm # head
>    speak:1_wayne_0_103_103,$107552$196064 # 9.722 # 15.254   # left arm # right arm # head
>    ```
>    In this timeline, a conflict arises between the upper body motion cues from "waves hands" (text description) and the speech audio during the overlapping period of 2.578 seconds to 8.0 seconds. However, **if users wish to control upper body motions using text during speech**, we can simply adjust the timeline as follows to achieve this:
>
>    ```
>    a man walks # 0.0 # 8.0 # legs # spine
>    wave hands # 7.0 # 11.0 # left arm # right arm
>    speak:1_wayne_0_103_103,$9248$99808 # 2.578 # 8.238   # left arm # right arm # head
>    speak:1_wayne_0_103_103,$107552$196064 # 9.722 # 15.254   # left arm # right arm # head
>    ```
>
> 	By modifying the timeline, the text command "wave hands" now controls the arms from 7.0 to 11.0 seconds. Since the speech audio spans from 2.578 to 15.254 seconds, there is an overlap between 7.0 and 11.0 seconds where both audio and text attempt to control the arms. According to our conflict resolution rules, the condition with fewer control parts takes precedence. In this case, the text command "wave hands" governs the arms during the overlapping period.
>
> 	As a result, users can control upper body motions using text while speech audio is active. The corresponding visualization is provided in the supplementary materials as **flexibility.mp4**.
>
> 4. **Visualization Results**
>
>    Thank you for your valuable advice. As suggested, we have added more video samples to the supplementary material in the **Qualitative Results** folder to visually demonstrate MOCO's performance.
>
> 5. **Real-World Testing.**
>
> 	Thank you for your valuable advice. As suggested, we have conducted experiments beyond controlled benchmarks to demonstrate how MOCO performs in real-world and diverse environments. Specifically, in more challenging scenarios, we utilized six different LLMs to generate text descriptions and time configurations to construct timelines based on the given audio content. We then used MOCO to generate motions according to these timelines. The visualization results are presented in the **LLM-boost-semantic** folder within the supplementary materials.
>
> 	Additionally, we tested MOCO's robustness with in-the-wild audio. We sampled two audio clips from SHOW, a speech-to-gesture dataset collected from real-world talk show videos, and used MOCO to generate motions based on these audio clips. The visualization results for these tests are available in the **generalization_to_show** folder within the supplementary materials.
>
> 	We observed that MOCO can generally generate realistic and semantically aligned motions based on the conditions automatically generated by LLMs. Furthermore, MOCO demonstrates robustness against out-of-domain audio inputs. Overall, MOCO exhibits superior performance in real-world and diverse environments.

---

> ### Author Response · Authors · 2024-12-03
>
> Thank you very much for taking the time to review our manuscript. We are pleased to hear that our previous responses have addressed most of your questions. Below, we provide our responses to your two additional concerns:
>
> 1. **Performance on Speech-to-Gesture Generation**
>
> 	Thanks for this concern. We would like to clarify that our joint generation architecture does not promote the performance of MOCO when it has single-modality condition. This is because our text-to-motion, speech-to-gesture, and speech-to-details denoisers are trained separately, which means experiments conducted on the speech-to-gesture dataset actually reflect the performance of the speech-to-gesture and speech-to-details denoisers only. In this setting, our joint generation architecture actually reduces to an ordinary speech-to-gesture generation model.
>
>  	Besides, MOCO indeed outperforms TalkShow and EMAGE when they have single-modality condition as input. One main reason is that TalkShow and EMAGE are based on VQ-VAE, while MOCO is diffusion-based. Diffusion models have been widely proven to be effective in various generation tasks. Therefore, we believe the advantage of MOCO compared with TalkShow and EMAGE are expected.
>
>
> 2. **Consistency between Facial Movements and Speech**
>
> 	Thanks for this concern. Our method achieves superior consistency between facial movements, particularly jaw and lip motions, and speech, which we have demonstrated through quantitative results. In **Table R2**, MOCO outperforms most methods, including advanced approaches for generating whole-body motion, including facial movements, such as TalkShow, EMAGE, and ProbTalk, on facial metrics. Specifically, MOCO achieves lower facial **MSE**, which measures the mean squared error between the ground truth and the generated facial vertices, and a facial **LVD** second only to EMAGE, which represents the velocity difference between the ground truth and the generated facial vertices. These performance metrics quantitatively illustrate the consistency and accuracy of our generated facial movements in relation to the corresponding speech.
>
> 	| Methods                         | FGD↓  | BC    | Diversity↑ | MSE↓  | LVD↓  |
> 	|---------------------------------|-------|-------|------------|-------|-------|
> 	| FaceFormer (2022)                     | -     | -     | -          | 7.787 | 7.593 |
> 	| CodeTalker (2023)                    | -     | -     | -          | 8.026 | 7.766 |
> 	| S2G (2019)                             | 28.15 | 4.683 | 5.971      | -     | -     |
> 	| Trimodal (2021)                       | 12.41 | 5.933 | 7.724      | -     | -     |
> 	| HA2G (2022)                           | 12.32 | 6.779 | 8.626      | -     | -     |
> 	| DisCo (2022)                        | 9.417 | 6.439 | 9.912      | -     | -     |
> 	| CaMN (2022)                          | 6.644 | 6.769 | 10.86      | -     | -     |
> 	| DiffStyleGesture (2023)                | 8.811 | 7.241 | 11.49      | -     | -     |
> 	| TalkShow (2023)                      | 6.209 | 6.947 | 13.47      | 7.791 | 7.771 |
> 	| EMAGE (2024)                     | 5.512 | 7.724 | 13.06      | 7.680 | 7.556 |
> 	| ProbTalk (2024)                  | 6.170 | 8.099 | 10.43      | 8.990 | 8.385 |
> 	| **MOCO (Ours)**                  | 5.543 | 7.089 | 14.05      | **7.285** | **7.573** |
>
> 	**Table R2**: Quantitative results of speech-to-gesture generation on the BEATX test set.
>
> 	Our qualitative results primarily focus on demonstrating full-body coordination, text-following, and broader application value, as suggested by reviewers eouV and FZJK. The emphasis on whole-body presentations makes it challenging to accurately assess small regions, such as facial expressions. As the deadline for uploading the revised PDF and supplementary materials has passed, we are unable to provide additional qualitative results to better address your concerns. However, we recommend viewing the video titled **generalization_to_show/show.mp4** in the supplementary. This video has a relatively higher resolution, and you can observe the synchronization of lip movements with speech.
>
> 	Furthermore, we are committed to including additional qualitative results and analyses of facial movements in the revised version.

---

### Official Review · Reviewer_eWKV · 2024-11-11

**Soundness:** 3
**Presentation:** 3
**Contribution:** 3
**Rating:** 5
**Confidence:** 4

**Summary:**

This paper proposes MOCO, a diffusion-based framework for generating coherent motions from multi-modal inputs like text, speech, and trajectory data. It addresses the challenge of simultaneous multi-modal control in 3D avatar motion generation. The key innovation is a decoupled denoising process that generates and combines motions for each modality. Experiments on a custom benchmark show it outperforms baselines.

**Strengths:**

(1) MOCO is quite straightforward and achieves better realism and coherence compared to existing methods that process modalities through weighted averaging.
(2) Comprehensive evaluation: The use of a purpose-built multi-modal benchmark and a wide range of metrics for evaluation provides a thorough assessment of the method's performance.
(3) The writing of paper is clear without ambiguities.

**Weaknesses:**

(1) While the paper mentions trajectory control, the evaluation focuses primarily on text-to-motion and speech-to-gesture tasks. A more comprehensive evaluation of trajectory control, including metrics specific to this modality, would be beneficial.

(2) The statistics of Multi-Modal Benchmark are missing.

(3) A Qualitative comparion among different methods (like User Study) are missing.

(4) The results of many evaluation metrics  are not promising at all, also authors only provided a cherry-picked video as demo, the effectiveness of this method is not so convincing.

(5)  Lacking theorical analysis of how decoupled-then-combined denoising works. Also, I am curious whethe is a general paradigm for multi-modal condition generation or only for this setting.

**Questions:**

(1) How about the performance comparison between "Synchronous" and  "Asynchronous" conditions?

(2) How many clips does your methods support to generate?(each one might be conditioned on several condition), and how the performance changes with the number of clips. Because your Figure 1. and demo video show you support this feature.

---

> ### Author Response · Authors · 2024-11-26
>
> 1. **Evaluation of Trajectory Control**
>    Thank you for your valuable advice. As suggested, we have conducted experiments to evaluate trajectory control on both location (measured in meters) and orientation (measured in radians), and the results are presented in the following table:
>
>    |        |     | Location | Location | Orientation | Orientation |
>    |------------------|-----------|-----------|--------------------|-----------------|--------------------|
>    | Classifier-Free Guidance (CFG) | L-BFGS    | Average Difference | Goal Difference | Average Difference | Goal Difference |
>    | ✘        | ✘ | 0.5641    | 1.2068             | 0.7059          | 1.2583             |
>    | ✔       | ✘ | 0.5676    | 1.3177             | 0.8276          | 1.5115             |
>    | ✘        | ✔| **0.0747**| **0.1235**         | **0.6009**      | **1.1031**         |
>    | ✔       | ✔| 0.1121    | 0.1950             | 0.7111          | 1.2845             |
>
>    **Table R1**: Evaluation of Trajectory Control.
>
>    For each category, we report two primary metrics:
>    - **Average Difference**: This metric quantifies the mean deviation between the generated trajectory and the ground truth (GT).
>    - **Goal Difference**: This metric measures the discrepancy at the final point relative to the GT.
>
>    The results indicate that L-BFGS optimization significantly reduces location differences and modestly enhances orientation accuracy. It is important to note that, for a given trajectory, orientation can vary widely, and thus the generated orientation does not need to closely match the GT. Conversely, incorporating CFG does not improve trajectory accuracy. These findings suggest that L-BFGS is an effective optimization strategy for trajectory control, and CFG may not offer additional benefits and could potentially disrupt the optimization process.
>
>    Additionally, we have included this analysis in **Appendix E.1**.
>
> 2. **Statistics of Multi-Modal Benchmark**
>    Thank you for your valuable advice. Originally, we outlined the multi-modal benchmark in **Section 4.1**. As suggested, we have added more statistical details of the benchmark along with the description of data's auto-generation pipeline. We have included this part in **Appendix F**.
>
> 3. **User Study**
>
> 	Thank you for your valuable advice.  As suggested, we have conducted user study and the results are presented in the below table.
>
> 	| **Method**          | **Better Text Following (%)** | **Better Beat Synchronization (%)** |
> 	|---------------------|-------------------------------|--------------------------------------|
> 	| Neither             | 1.0                           | 13.0                                 |
> 	| Pseudo-Text         | 0.0                           | 12.5                                 |
> 	| MOCO (Ours)         | 99.0                          | 74.5                                 |
>
> 	| **Method**          | **Better Text Following (%)** | **Better Beat Synchronization (%)** |
> 	|---------------------|-------------------------------|--------------------------------------|
> 	| Neither             | 0.0                           | 13.5                                 |
> 	| Weighted Average    | 0.0                           | 3.5                                  |
> 	| MOCO (Ours)         | 100.0                         | 83.0                                 |
>
> 	| **Method**               | **Better Body Coherence (%)** | **Better Temporal Fluidity (%)** |
> 	|--------------------------|-------------------------------|-----------------------------------|
> 	| Neither                  | 12.0                          | 12.6                              |
> 	| Combine Only Last Time   | 17.0                          | 42.6                              |
> 	| MOCO (Ours)              | 71.0                          | 44.8                              |
>
> 	Specifically, we evaluate MOCO against *Pseudo-Text* and *Weighted Average*, introduced in Section 4.3, to assess overall performance in text following and audio beat synchronization. Additionally, we compare MOCO with *Combine Only Last Time*, which also employs the decoupling strategy but applies the combining strategy only at the final diffusion step. This comparison aims to evaluate whole body coherence and temporal fluidity.
>
> 	As shown in the table, MOCO achieves significant advantages over both *Pseudo-Text* and *Weighted Average*, demonstrating the effectiveness of our decouple-then-combine strategy in generating motion aligned with multi-modal conditions. Furthermore, when compared to *Combine Only Last Time*, our method was rated significantly higher in both body coherence and temporal fluidity. This indicates that MOCO does more than merely combine different body parts controlled by separate conditions; it ensures that each body part aligns with its corresponding condition while enhancing coordination among all body parts.

---

> ### Author Response · Authors · 2024-11-26
>
> 4. **Visualization Results**
>
>     Thank you for your valuable advice. As suggested, we have added more video samples to the supplementary material in the **Qualitative Results** folder to visually demonstrate MOCO's performance.
>
> 5. **Evaluation Metrics**
>
>    Thank you for your valuable advice. To accurately assess the effectiveness of our method, it is essential to consider metrics across all relevant domains simultaneously, including both text-to-motion and speech-to-gesture. When evaluated from this comprehensive perspective, MOCO achieves the best overall performance compared to baseline methods. We acknowledge that many evaluation metrics may not initially appear promising in each single-modality condition. This is because MOCO generates motions by integrating both text-driven motion and speech-driven gestures, resulting in a distribution that significantly differs from the ground truth (GT) in either text-to-motion or speech-to-gesture datasets alone. Since most evaluation metrics are calculated based on the differences between the generated results and the GT, MOCO's performance on individual domain metrics may not seem superior.
>
>    To better demonstrate the strengths of our approach, we have included additional comparative videos in the supplementary materials. Additionally, a user study is currently underway, and we will update the results as soon as they are available.
>
> 6. **Theoretical Analysis**
>
> 	Thank you for your valuable advice. As suggested, we have added a theoretical analysis in **Appendix A**. Below is a summary of the key points:
>
> 	Our "decoupled-then-combined" strategy is based on approximating the joint conditional probability $p(x_{t-1} \mid c_{\text{text}}, c_{\text{audio}}, x_t)$ by **decoupling** it into two independent probabilities and then **combining** them, which can be expressed as:
>
> 	$$
> 	p(x_{t-1} \mid c_{\text{text}}, c_{\text{audio}}, x_t) \approx p(x_{t-1, \text{lower}} \mid c_{\text{text}}, x_t) \times p(x_{t-1, \text{upper}} \mid c_{\text{audio}}, x_t)
> 	$$
>
> 	**Underlying Assumptions**:
> 	- Information Sufficiency: The current state $x_t$ contains enough information about $x_{t-1}$ to allow the decoupling without significant loss of accuracy. This is justified by the proximity of diffusion steps and the high correlation between consecutive states.
> 	- Modality-Specific Influence: Text input $c_\text{text}$  primarily affects lower-body movements (e.g., walking, stance shifts), while audio input $c_\text{audio}$  predominantly influences upper-body movements (e.g., gestures, facial expressions).
>
> 	For a more detailed explanation and mathematical derivations, please refer to **Appendix A**.
>
> - **General paradigm for multi-modal condition generation**. Although our "decoupled-then-combined" approach is designed to address the interactions between text and audio modalities in the context of motion generation, its underlying principle of decoupling and subsequently combining conditional probabilities has broader potential. As long as the assumptions outlined in Appendix A are satisfied, this approach could be extended to other multi-modal settings. The general applicability, however, depends on the nature of the modalities and their interactions in the target task. Future research could investigate how this paradigm can be adapted to various combinations of modalities and domains.

---

> ### Author Response · Authors · 2024-11-27
>
> 7. **Comparison between "Synchronous" and "Asynchronous" Conditions**
>    Thank you for your valuable advice. As suggested, we have conducted experiments under both synchronous and asynchronous conditions, with the results presented in the following table.
>
>    |                           | FID+ $\downarrow$  | R1 $\uparrow$  | R3 $\uparrow$  | M2T $\uparrow$  | M2M $\uparrow$  | FID-A $\downarrow$  | BC $\uparrow$  | L1div $\uparrow$  | MTD $\downarrow$ |
>    |---------------------------|--------------------|----------------|----------------|-----------------|-----------------|---------------------|----------------|-------------------|------------------|
>    | Ground Truth              | 0.000              | 40.0           | 72.5           | 0.781           | 1.000           | -                   | -              | -                 | 2.9              |
>    | Synchronous               | 0.896              | 23.8           | 45.9           | 0.638           | 0.629           | 4.41                | 2.62           | 9.47              | 5.5              |
>    | Asynchronous              | 0.862              | 24.6           | 46.9           | 0.649           | 0.639           | 3.83                | 2.72           | 8.62              | 5.3              |
>
> 	**Table R2**: Comparison of MOCO in synchronous and asynchronous conditions.
>
> 	As presented in the table, MOCO generates slightly superior motions under asynchronous conditions compared to synchronous ones. This improvement may stem from asynchronous conditions allowing a single modality to control the entire body, which is potentially less complex than using multiple modalities to simultaneously control different parts.
>
>    We have included this analysis in **Appendix E.2**.
>
> 8. **Clips Our Method Supports to Generate**
>
>     Thanks to the relatively lightweight network architecture and effective multi-condition management, our method can simultaneously generate hundreds of clips with different conditions.

---

### Comment · Reviewer_eouV · 2024-11-25

Dear authors, it seems the deadline of communication period is coming. Do you have any update? Thank you!

---

> ### Author Response · Authors · 2024-11-25
>
> Thank you very much for the kind reminder. We will respond to all the reviewers simultaneously to demonstrate our respect. As addressing the valuable feedback from all the reviewers requires considerable efforts, we will update our responses tomorrow. We sincerely appreciate your patience and great support!

---

### Author Response · Authors · 2024-12-04
**Summary**

### **Dear Chairs and Reviewers**:

First and foremost, we would like to express our sincere gratitude for your time and valuable insights, which have greatly contributed to improving our paper. Below, we provide a summary of the strengths highlighted by each reviewer, along with our discussions with them, to help the chairs quickly grasp the key points of the discussion.

### **Strengths**

**TLDR** (summarized from reviewers' statements about strengths):

> MOCO introduces **"the first approach"** to handle simultaneous multi-modal inputs (speech, text, and trajectory) in motion synthesis, **"addressing a significant gap"**  in existing research and resulting in **"realistic and synchronized avatar behavior"**. The purpose-built multimodal benchmark is **"well-considered"**, and the experiments are **"thorough."** The paper is **"well-structured and clearly articulated."** and the problem it tackles has **"important implications"** for virtual avatars, gaming, and animation industries.

The reviewers have identified several strengths of our paper, as outlined below:

- **Novelty** [rsd5, itKM, FZJK]:
  Reviewer rsd5 highlights that MOCO introduces the first approach for handling simultaneous multi-modal inputs (speech, text, and trajectory). Reviewer itKM describes the framework as "novel," while Reviewer FZJK notes that the approach is "innovative."

- **Effectiveness** [eWKV, rsd5, itKM, FZJK]:
  Reviewer eWKV commends the paper for "achieving better realism and coherence" compared to existing methods. Reviewer rsd5 adds that MOCO leads to "realistic and synchronized avatar behavior." Reviewer itKM appreciates the "well-designed decoupled denoising approach," which "enhances the coherence and synchronization of generated motions." Reviewers itKM and FZJK highlight the method’s effectiveness in "addressing a significant gap" in the field.

- **Comprehensive Evaluation** [eWKV, eouV, FZJK]:
  Reviewer eWKV notes that the paper provides a "thorough assessment" of the proposed method. Reviewer eouV emphasizes that "the experiments are thorough," "the benchmark is well-considered," and "the ablation study is comprehensive." Reviewer FZJK states that our benchmark "enhances the validity" of our claims.

- **Clarity and Quality of Presentation** [eWKV, itKM, eouV, FZJK]:
  The clarity and quality of the presentation were consistently praised by Reviewers eWKV, itKM, eouV, and FZJK.

- **Relevance and Significance** [itKM, eouV]:
  Reviewer itKM emphasizes the method’s importance for synthesizing motion from multi-modal inputs, noting its importance to "virtual avatars, gaming, and animation industries." Reviewer eouV also acknowledges the problem is "interesting and has application value.""

### **Discussion**

**Improved Rating**

After discussion, **Reviewer eouV** stated, **"I feel a large part of my concern has been addressed,"** and improved their rating from 5 to 6. The reviewer’s insight into using large language models (LLMs) for motion generation and editing is **of great value.** The corresponding results are **"interesting and inspiring, and they show the potential for broader and easier use of this method."** We sincerely thank Reviewer eouV for their thoughtful feedback.

Similarly, after discussion, **Reviewer FZJK** raised their rating from 6 to 8. The reviewer highlighted the potential value of addressing contradictory motion cues, which motivates us to further enhance the flexibility of our approach. Upon reviewing the rebuttal, the reviewer commented, **"the exploration of this work is valuable for the community and has great potential for application."** We are grateful to Reviewer FZJK for their constructive insights.

**Declined Rating**

After discussion, **Reviewer rsd5** stated that **"thank you to the authors for addressing most of my questions,"** and further raised two new concerns. Due to the low-resolution of the video demos, the assessment of consistency between facial movements and speech becomes difficult and therefore forms the main concern. The reviewer temporarily lowered their rating from 6 to 5. In response, we present **quantitative results to clarify that our method achieves strong performance compared to advanced methods.** We also provide a high-resolution video demo to clarify that our model is superior in the consistency between facial movements and speech. We look forward to further feedbacks from the reviewer.

**No Further Reply Yet**

We have addressed each of the concerns raised by the other reviewers in the corresponding sections of the official review and look forward to their feedback.

---

### Meta-Review · Area_Chair_uvDX · 2024-12-20

**Metareview:**

The submission is about multi-modal motion synthesis.  Reviewers acknowledged that the submission introduces a new framework; however, they had concerns regarding the technical novelty of the solution (as it seems to be an integration of existing modules), as well as the practical applicability of the proposed new task.  Post rebuttal, some reviewers became more positive, but the overall sentiment has not changed.  The AC agreed with the concerns and recommended the authors revise the submission thoroughly for the next venue.

**Additional Comments On Reviewer Discussion:**

The rebuttal was effective in helping some reviewers change their opinion, though the overall sentiment has not changed.

---

### Decision · Program_Chairs · 2025-01-22

Reject